# Novel ATP-cone-driven allosteric regulation of ribonucleotide reductase via the radical-generating subunit

Inna Rozman Grinberg[1], Daniel Lundin[1], Mahmudul Hasan[1,2†], Mikael Crona[3†], Venkateswara Rao Jonna[4†], Christoph Loderer[1], Margareta Sahlin[1], Natalia Markova[5], Ilya Borovok[6], Gustav Berggren[7], Anders Hofer[4], Derek T Logan[2]*, Britt-Marie Sjöberg[1]*

[1]Department of Biochemistry and Biophysics, Stockholm University, Stockholm, Sweden; [2]Department of Biochemistry and Structural Biology, Lund University, Lund, Sweden; [3]Swedish Orphan Biovitrum AB, Stockholm, Sweden; [4]Department of Medical Biochemistry and Biophysics, Umeå University, Umeå, Sweden; [5]Malvern Instruments Inc., Sweden; [6]Department of Molecular Microbiology and Biotechnology, Tel-Aviv University, Tel Aviv-Yafo, Israel; [7]Department of Chemistry, Uppsala University, Uppsala, Sweden

*For correspondence:
derek.logan@biochemistry.lu.se (DTL);
britt-marie.sjoberg@dbb.su.se (B-MSö)

†These authors contributed equally to this work

Competing interests: The authors declare that no competing interests exist.

**Abstract** Ribonucleotide reductases (RNRs) are key enzymes in DNA metabolism, with allosteric mechanisms controlling substrate specificity and overall activity. In RNRs, the activity master-switch, the ATP-cone, has been found exclusively in the catalytic subunit. In two class I RNR subclasses whose catalytic subunit lacks the ATP-cone, we discovered ATP-cones in the radical-generating subunit. The ATP-cone in the *Leeuwenhoekiella blandensis* radical-generating subunit regulates activity via quaternary structure induced by binding of nucleotides. ATP induces enzymatically competent dimers, whereas dATP induces non-productive tetramers, resulting in different holoenzymes. The tetramer forms by interactions between ATP-cones, shown by a 2.45 Å crystal structure. We also present evidence for an $Mn^{III}Mn^{IV}$ metal center. In summary, lack of an ATP-cone domain in the catalytic subunit was compensated by transfer of the domain to the radical-generating subunit. To our knowledge, this represents the first observation of transfer of an allosteric domain between components of the same enzyme complex.
DOI: https://doi.org/10.7554/eLife.31529.001

## Introduction

Allosteric regulation of an enzyme is defined as regulation of activity by binding of an effector molecule to a different location of the enzyme than the active site. The effector influences the distribution of tertiary or quaternary conformations of an enzyme, alone or in combination, modulating its activity (*Swain and Gierasch, 2006*). Allostery is an intrinsic property of many, if not all, dynamic proteins (*Gunasekaran et al., 2004*) and an important factor in disease (*Nussinov and Tsai, 2013*), and has hence attracted considerable scientific interest. A substantial part of this interest has been focused on ribonucleotide reductase (RNR), which has been termed a 'paradigm for allosteric regulation of enzymes' (*Aravind et al., 2000*).

RNRs are essential enzymes in all free-living cells, providing the only known de novo pathway for the biosynthesis of deoxyribonucleotides (dNTPs), the immediate precursors for DNA synthesis and repair (*Hofer et al., 2012*; *Nordlund and Reichard, 2006*). To avoid imbalanced levels of dNTPs and the increased mutation rates that are the inevitable consequences of this (*Kumar et al., 2011*; *Mathews, 2006*; *Watt et al., 2016*), RNRs are tightly controlled through transcriptional and

**eLife digest** When a cell copies its DNA, it uses four different building blocks called deoxyribonucleotides (dNTPs). These consist of one of the four 'bases' (A, T, C and G), which pair up to link the two strands of DNA in the double helix, bound to a sugar and a phosphate group. If the cell contains too little or too much of one of these building blocks, an incorrect base may be inserted into the DNA. This results in a mutation, which in bacteria can cause death, and in animals may lead to cancer.

The enzyme that fabricates and carefully controls the amount of each dNTP building block inside a cell is called ribonucleotide reductase. Once there are enough building blocks in a cell the enzyme is turned off. A part of the enzyme called the ATP-cone acts as an on/off switch to control this activity.

The ribonucleotide reductase consists of a large component and a small component. Until now, studies of the ATP-cone have found it only in the large component of the enzyme. However, when looking through a public database of sequence data, Rozman Grinberg et al. noticed that ribonucleotide reductases in some bacteria have their ATP-cone joined to the small component. Does this ATP-cone also control the amounts of dNTP building blocks inside cells and, if so, how?

Rozman Grinberg et al. studied one such ATP-cone in a ribonucleotide reductase from a bacterium (named *Leeuwenhoekiella blandensis*) found in the Mediterranean Sea. This revealed that when the amount of dNTP building blocks reaches a certain limit, the ATP-cone turns off the enzyme. Examining the three-dimensional structure of the enzyme using a technique called X-ray crystallography revealed that when turned off, the enzyme's small components are glued together in pairs. This prevents them from working. Rozman Grinberg et al. also discovered that this enzyme contains a new type of metal center with two manganese ions suggesting that a new reaction mechanism may operate in this class of ribonucleotide reductase.

These findings support a theory that biological on/off switches can evolve rapidly. In addition to its evolutionary and biomedical interest, understanding how the ATP-cone works might help to improve the enzymes used in industrial processes.

DOI: https://doi.org/10.7554/eLife.31529.002

allosteric regulation, subcellular compartmentalization and small protein inhibitors (*Pai and Kearsey, 2017*; *Torrents, 2014*). Allosteric regulation of RNRs affects both substrate specificity and overall activity. The specificity regulation has been intensively studied and described for all three classes of RNRs (*Andersson et al., 2000*; *Hofer et al., 2012*; *Larsson et al., 2004*; *Reichard, 2010*; *Torrents et al., 2000*; *Zimanyi et al., 2016*). The s-site binds dNTPs and determines which nucleotide will be reduced at the active site to ensure balanced levels of the four dNTPs in the cell. Additionally, many RNRs possess an overall activity regulation site (a-site) (*Brown and Reichard, 1969*; *Thelander and Reichard, 1979*) positioned in an N-terminal domain of ~85–100 amino acid residues called the ATP-cone (*Aravind et al., 2000*; *Eriksson et al., 1997*). Acting as a regulatory master switch, the a-site senses intracellular nucleotide concentrations by competitive binding of ATP and dATP. In the presence of ATP the enzyme is active, and when concentrations of dNTPs rise, binding of dATP switches the enzyme off. This mechanism ensures sufficient but not excessive amounts of nucleotides that may also cause increased mutation rates (*Mathews, 2006*).

The ATP-cone is an example of allosteric regulation controlled by a domain that acts relatively independent of the catalytic core of proteins. This type of allosteric regulation has been shown to provide an evolutionarily dynamic process by which allosteric regulation can be lost or gained both in RNRs (*Lundin et al., 2015*) and other enzymes (*Aravind and Koonin, 1999*; *Lang et al., 2014*). Although regulation has been lost and gained repeatedly in RNRs through evolutionary time, to date, the ATP-cone domain has been observed exclusively at the N-terminus of the catalytic subunit NrdA (class I), NrdJ (class II) and NrdD (class III). Class I RNRs consist of a large, catalytic subunit (α or NrdA), and a smaller, radical-generating subunit (β or NrdB) (*Huang et al., 2014*; *Nordlund and Reichard, 2006*). NrdA and NrdB interact to form the active complex, in which the two proteins need to be precisely positioned such that the radical can be transferred from NrdB, where it is generated and stored, to NrdA, where it starts the substrate reduction. In class I, it has long been noted

that ATP-cones are absent from subclass Ib (NrdE) but present in several, but not all, NrdAs. A recent phylogenetic subclassification of RNRs reveals that three phylogenetically well-supported subclasses of class I never have ATP-cones (*Jonna et al., 2015*) (http://rnrdb.pfitmap.org): NrdE, NrdAi and NrdAk. In two of these subclasses we instead discovered ATP-cones attached to their corresponding radical-generating subunit: NrdF (the Ib subclass) and NrdBi. It hence appears as if the lack of activity regulation through an ATP-cone in the catalytic subunit is compensated by the presence of this domain in the non-homologous radical-generating subunit of some RNRs.

Three distinct types of class I complexes have been mechanistically characterized and found to operate via nucleotide-induced regulation of quaternary structure (*Johansson et al., 2016*; *Jonna et al., 2015*; *Kashlan et al., 2002*; *Rofougaran et al., 2008*; *Rofougaran et al., 2006*; *Torrents et al., 2006*). Crystal structures, cryo-electron microscopy reconstructions and small-angle X-ray scattering studies of inhibited complexes have revealed that when dATP is bound at the a-site, high molecular mass oligomers are formed, in which the radical transfer pathway is distorted (*Ando et al., 2011*; *Ando et al., 2016*; *Fairman et al., 2011*; *Johansson et al., 2016*). Conversely, when ATP is bound, an active enzyme complex is formed. Interestingly, the structure and organization of subunits in active and inactive complexes varies considerably between species (*Ahmad and Dealwis, 2013*; *Hofer et al., 2012*). In *Escherichia coli* RNR, the active NrdAB complex is $\alpha_2\beta_2$, whereas the inactive form is an $\alpha_4\beta_4$ ring-shaped octamer where the ATP-cones in the $\alpha$ subunits sequester the $\beta$ subunits in a non-productive conformation (*Ando et al., 2011*). In the eukaryotic class I RNR, the inactive complex differs from the one in *E. coli* in that it has an $\alpha_6$ stoichiometry. This hexamer can only bind one $\beta_2$ subunit in an unproductive manner without a properly aligned electron transport chain (*Fairman et al., 2011*). Activation by ATP creates a different type of $\alpha_6$ complex that binds one or more $\beta_2$ complexes (*Ando et al., 2016*; *Aye and Stubbe, 2011*; *Crona et al., 2013*; *Fairman et al., 2011*; *Rofougaran et al., 2006*). The different complexes are formed by subtle changes at the a-site induced by binding of the different nucleotides (*Fairman et al., 2011*; *Xu et al., 2006*). Another oligomerization mechanism has been recently found in *Pseudomonas aeruginosa* class I RNR, which possesses two consecutive ATP-cones, of which only the N-terminal one binds nucleotides. The active complex is once again $\alpha_2\beta_2$, but the inactive *P. aeruginosa* RNR complex is a dATP-induced $\alpha_4$ complex consisting of a ring of four $\alpha$ subunits interacting via their outer ATP-cones (*Johansson et al., 2016*; *Jonna et al., 2015*). A single $\beta_2$ can bind to this ring, but the complex is inactive. Oligomerization of RNRs may be a useful character to explore biomedically. RNRs have long attracted interest as potential targets for novel antibiotics as well as for cancer therapy (*Aye et al., 2015*; *Julián et al., 2015*; *Tholander and Sjöberg, 2012*). Several RNR drugs are directed towards the radical-containing subunit and the active site of the catalytic subunit. Recently, some nucleoside analogs used in cancer treatments and known to inhibit RNRs in vitro in their phosphorylated forms were shown to induce hexameric complexes in vivo (*Aye et al., 2012*; *Aye and Stubbe, 2011*; *Wisitpitthaya et al., 2016*).

The unexpected finding of an ATP-cone fused to the radical-generating subunits poses questions regarding how it might regulate activity. Here we describe the mechanism of activity regulation by the ATP-cone N-terminally fused to the radical-generating NrdBi from *Leeuwenhoekiella blandensis* sp. nov. strain MED217. *L. blandensis* was isolated from Mediterranean surface seawater and belongs to Flavobacteriaceae, the major family of marine Bacteroidetes, with important roles in carbon flow and nutrient turnover in the sea during and following algal blooms (*Fernández-Gómez et al., 2013*; *Pinhassi et al., 2006*). *L. blandensis* possesses two RNRs: a class II without ATP-cone, and the class I NrdAi/NrdBi, which lacks an ATP-cone in NrdA and instead contains an ATP-cone positioned at the N-terminus of NrdB. Superficially, the allosteric mechanism of *L. blandensis* NrdAi/NrdBi holoenzyme is similar to when the ATP-cone is contained in the catalytic subunit. At high dATP concentrations, inhibited higher oligomeric complexes of the holoenzyme are favoured. However, in the *L. blandensis* class I RNR, the oligomerization is driven by a shift towards tetramers of the radical-generating subunit. This illustrates how allosteric regulation controlled by ATP-cones can evolve in a highly dynamic way, requiring few adaptations to the core of the enzyme. The relative ease by which ATP-cone-driven activity regulation appears to evolve, provides a potential route to regulate engineered enzymes by dATP-inhibition for enzymes in which activity is affected by oligomerization. Addition of an ATP-cone to the protein could be used to induce higher oligomers controlled by dATP addition.

## Results

### The activity allosteric regulatory ATP-cone is linked to the radical-containing subunit in some class I RNRs

We detected ATP-cones in the radical-generating subunits of RNRs from two distinct phylogenetic RNR subclasses: NrdBi and NrdF (*Figure 1a*). In the NrdBi sequences, the ATP-cone was found at the N-terminus of the protein, whereas it was found at the C-terminus of the NrdF proteins (*Figure 1b*). Interestingly, the corresponding catalytic subunit subclasses – NrdAi and NrdE respectively – have been found to lack ATP-cones. Ninety-three sequences in NCBI's RefSeq database are NrdBi proteins with N-terminally positioned ATP-cones. They are encoded by viruses and bacteria from several phyla, although the Flavobacteriales order in the Bacteroidetes phylum predominate (70 sequences, http://rnrdb.pfitmap.org). NrdF proteins with a C-terminally positioned ATP-cone are only encoded by a few species of the Meiothermus genus of the Deinococcus-Thermus phylum (http://rnrdb.pfitmap.org). All species encoding NrdB proteins with ATP-cones in their genomes also encode other RNRs.

### Substrate specificity regulation of *L. blandensis* RNR via the s-site

Initially we cloned, expressed and purified the *L. blandensis* NrdBi and NrdAi proteins. Using a four-substrate activity assay in the presence of saturating concentrations of the s-site effectors dTTP, dGTP or ATP, we found that *L. blandensis* RNR has a similar specificity regulation pattern to most characterized RNRs (*Hofer et al., 2012*). ATP stimulated the reduction of CDP and UDP, whereas dTTP stimulated the reduction of GDP, and dGTP stimulated the reduction of ADP and GDP (*Figure 2*). The enzyme was completely inactive in the absence of allosteric effectors. Using mixtures of allosteric effectors, we observed that dTTP-induced GDP reduction dramatically increased in the presence of ATP (*Figure 2*).

### Overall activity of *L. Blandensis* RNR is regulated via the NrdB-linked ATP-cone

We performed a series of activity assays with CDP as substrate to elucidate the potential roles of ATP and dATP in activating and inhibiting the enzyme (*Figure 3*). The presence of ATP clearly activated the enzyme (*Figure 3a*), while dATP activated the enzyme when used at low concentrations and was inhibitory at 30 µM and higher (*Figure 3b*). An ATP-cone deletion mutant NrdBΔ99, lacking the N-terminal 99 residues, reached a lower maximum activity compared to the wild type enzyme, suggesting that it was not activated by ATP beyond saturation of the s-site in the NrdA, nor was it inhibited by dATP (*Figure 3a–b*). From the NrdBΔ99 effector titrations, we could calculate $K_L$ values – the concentrations of allosteric effectors that give half maximal enzyme activity – for binding of ATP and dATP to the s-site in NrdA to 30 and 1.4 µM respectively. Titration of ATP into wild type NrdB in the presence of an excess of NrdA saturated with dTTP showed that it activates the enzyme through the a-site with a $K_L$ of 96 µM (*Figure 3c*). For the corresponding inhibition by dATP binding to the a-site, we calculated the $K_i$ value - the binding constant of a non-competitive inhibitor - to be 20 µM in the presence of an s-site saturating dTTP concentration (*Figure 3d*). We also tested if only the triphosphate form of (deoxy)adenosine nucleotides would interact with the a-site in presence of s-site saturating dTTP concentrations. In addition to ATP and dATP, dADP was also found to interact, whereas ADP, AMP and dAMP had no effect (*Figure 3e*). Titration with dADP inhibited the enzyme activity, although less strongly than dATP (*Figure 3f*).

### dATP binding to NrdB induces formation of higher oligomeric complexes

To elucidate the mechanism of allosteric overall activity regulation governed by the NrdB-linked ATP-cone, activity-independent oligomer-distribution experiments were performed by gas-phase electrophoretic macromolecule analysis (GEMMA). In the absence of allosteric effectors, NrdB (β) is mainly monomeric (theoretically 51.8 kDa) and, in contrast to most studied NrdB proteins, it does not readily form dimers at the low protein concentration range suitable for GEMMA analyses (*Figure 4a*). Addition of dATP (50 µM) dramatically shifted the equilibrium towards tetramers $\beta_4$, which became the major form under these conditions (*Figure 4a*). Titration of increasing

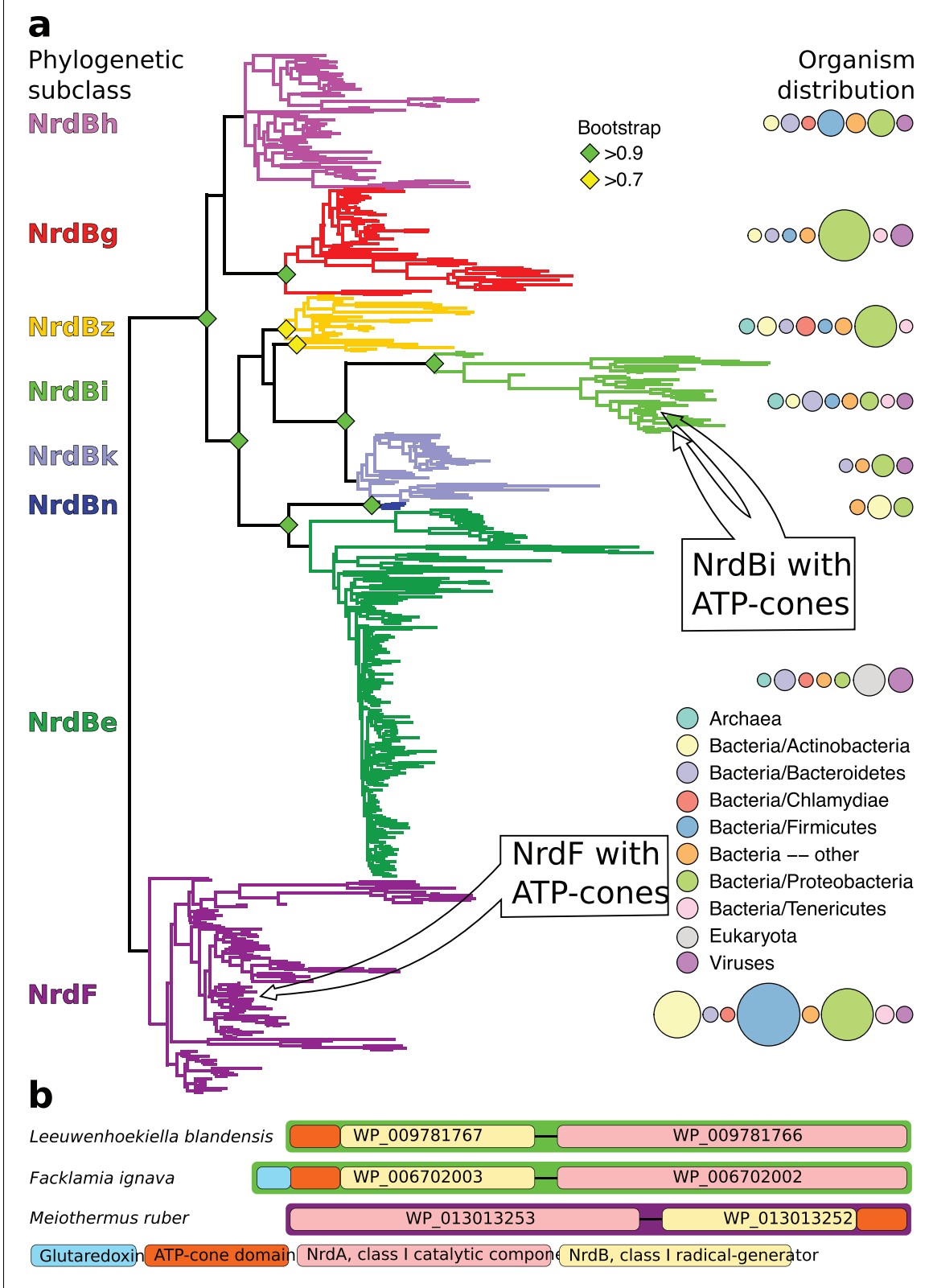

**Figure 1.** Unrooted phylogenetic tree of NrdB sequences and genome arrangements of NrdB sequences with ATP-cones. (**a**) Maximum likelihood phylogeny of class I RNR radical generating subunit with subclasses in color, names to the left and organism distributions to the right (see inset legend; sizes of circles are proportional to the number of sequences found in each taxon). Bootstrap support values greater than 0.7 are shown with colored diamonds. NrdBs with ATP-cones were discovered in two subclasses: NrdBi and NrdF (formerly class Ib). Neither of the two subclasses have

*Figure 1 continued on next page*

*Figure 1 continued*

corresponding catalytic subunits, NrdAi and NrdE respectively, with ATP-cones. In both subclasses, NrdB sequences with ATP-cones were rare and phylogenetically limited, see inset arrows. (b) Arrangement of class I RNR genes in three genomes encoding NrdB proteins with ATP-cones (*green borders*, NrdAi/NrdBi; *purple borders*, NrdE/NrdF). Genes are shown 5' to 3', so that ATP-cones in the N-terminus are to the left in the gene.

DOI: https://doi.org/10.7554/eLife.31529.003

concentrations of dATP to NrdB showed that the tetramers reached their half-maximum mass concentration at around 10 µM dATP (*Figure 4—figure supplement 1*). Addition of 50 µM dADP also induced formation of NrdB tetramers, although less efficiently than with dATP (*Figure 4a*). In contrast, the NrdBΔ99 mutant, lacking the ATP-cone (*Figure 4b*), was mainly monomeric regardless if dATP was present or not (*Figure 4b*), demonstrating that the NrdB-linked ATP-cone is required for dATP-induced tetramer formation. The protein had a tendency to aggregate, which can possibly explain the ladder of monomers, dimers, trimers and tetramers. NrdA was a monomer (theoretically 70.6 kDa) in the absence of allosteric effectors, while addition of dATP prompted formation of dimers (*Figure 4c*) To assess the oligomeric state of the complete enzymatic complex of the inactive *L. blandensis* RNR, a mixture of NrdA (α) and NrdB (β) in the presence of 100 µM dATP was analyzed with GEMMA (*Figure 4c*). The two subunits formed a large complex of 340 kDa with the expected mass of an $\alpha_2\beta_4$ complex (theoretically 348 kDa), and at higher protein concentration a complex of 465 kDa with the expected mass of an $\alpha_4\beta_4$ complex (theoretically 488 kDa) started to appear.

To complement the GEMMA analyses of oligomer formation, we performed analytical size exclusion chromatography (SEC) using higher protein concentrations and physiologically reasonable concentrations of effectors (3 mM ATP and 0.1 to 0.5 mM dATP) (*Bochner and Ames, 1982*; *Buckstein et al., 2008*) on an analytical Superdex 200 PC 3.2/30 column at 7°C (*Figure 5*). These experiments were complemented with SEC experiments at room temperature with a semi-preparative column (Superdex 200 10/300 GL) that gave better resolution but was less practical for the bulk experiments due to its much higher mobile phase consumption. The SEC experiments showed that

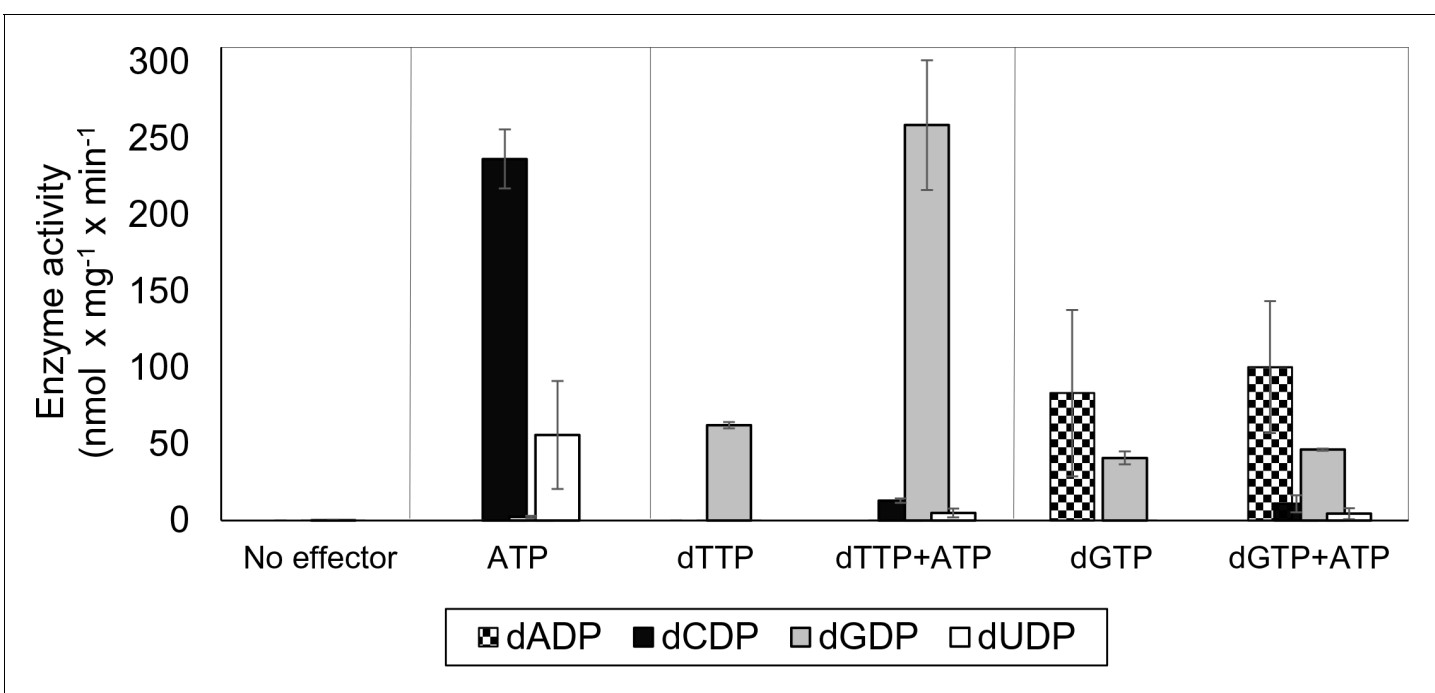

**Figure 2.** Allosteric specificity regulation of the *L. blandensis* class I RNR. Enzyme activity was measured after 10 and 30 min in the presence of all four substrates (0.5 mM each) and with the indicated allosteric effectors (2 mM of each). Error bars indicate the extremes of two measurements. Protein concentrations were 1 µM NrdB and 4 µM NrdA.

DOI: https://doi.org/10.7554/eLife.31529.004

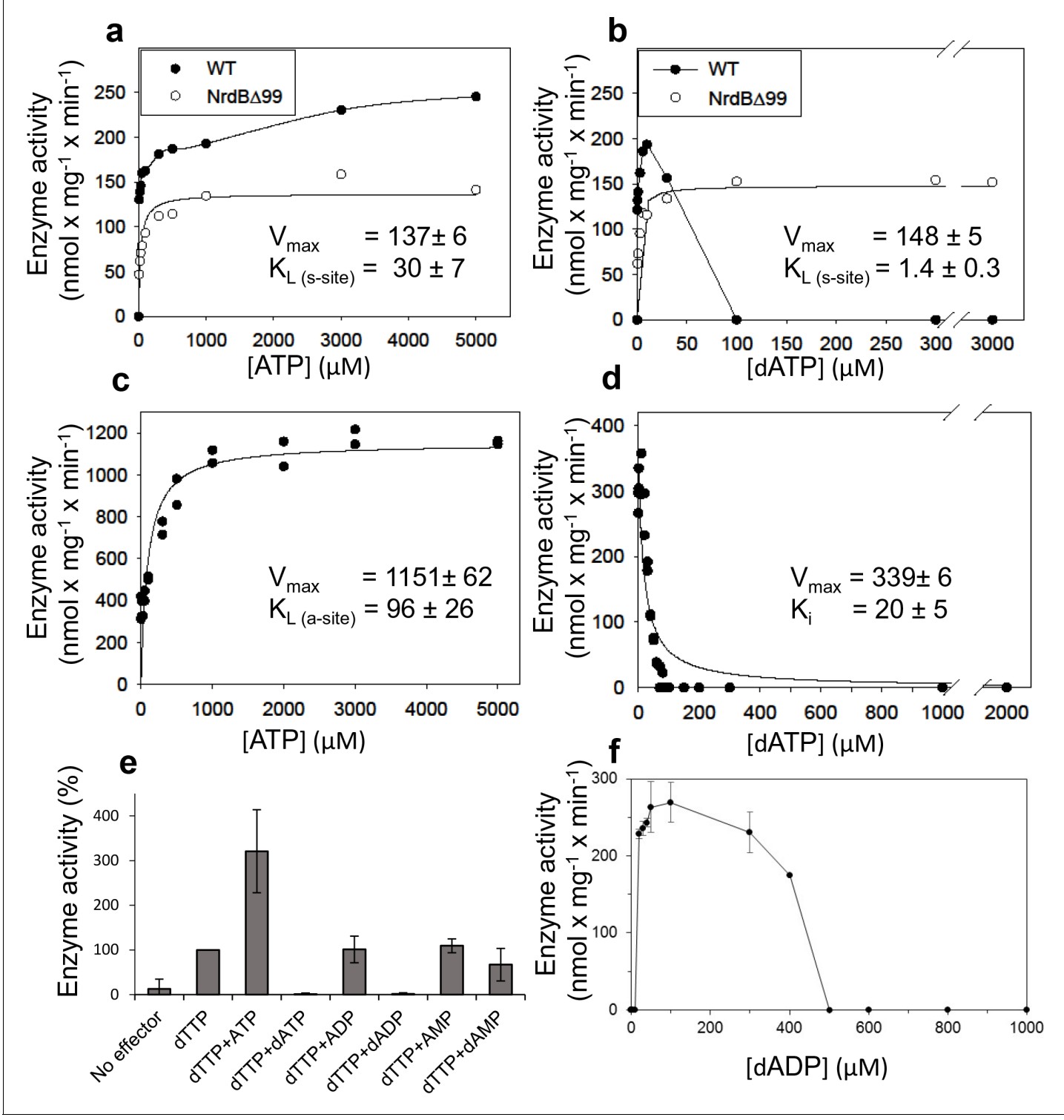

**Figure 3.** Activity of *L. blandensis* RNR with wild type NrdB or NrdBΔ99 in the presence of allosteric effectors. NrdB was used in excess over NrdA when studying the s-site and NrdA was used in excess when studying the a-site. (**a–b**) CDP reduction assayed in mixtures of 0.5 µM NrdA and 2 µM of either wild type NrdB (black circles) or NrdBΔ99 (white circles), titrated with ATP (**a**) or dATP (**b**). (**c**) GDP reduction assay mixtures with 2 µM NrdA and 0.5 µM wild type NrdB titrated with ATP in the presence of an s-site saturating concentration of dTTP (2 mM). (**d**) Reduction of GDP assayed with 2 µM NrdA and 0.5 µM wild type NrdB, titrated with dATP in the presence of an s-site saturating concentration of dTTP (2 mM). (**e**) GDP reduction in presence of s-site saturating dTTP (2 mM) and 2 mM of the indicated adenosine nucleotides. Assay mixtures contained 4 µM NrdA and 1 µM NrdB.

*Figure 3 continued on next page*

*Figure 3 continued*

100% activity corresponded to 639 nmol x mg$^{-1}$ x min$^{-1}$. (**f**) CDP reduction assays titrated with dADP. Assay mixtures contained 2 μM NrdA and 0.5 μM NrdB. Error bars in panels E and F indicate the standard deviation of three measurements.

DOI: https://doi.org/10.7554/eLife.31529.005

NrdB was in a monomer-dimer-tetramer equilibrium (*Figure 5—figure supplement 1a*) with an average size comparable to dimers (*Figure 5a*). Similar to the results with GEMMA, it was mainly a monomer at lower protein concentrations (*Figure 5—figure supplement 1a*). ATP and dATP promoted the equilibrium to shift to dimers and tetramers, respectively (*Figure 5a*, *Figure 5—figure supplement 1b*). Without effectors, NrdA was mainly in a monomeric state, while it was dominated by dimers in the presence of either of the two effectors (*Figure 5b*). However, in agreement with the GEMMA results the formation of the nucleotide-induced dimer was much less efficient at lower protein concentration (*Figure 5—figure supplement 1c*). When NrdA and NrdB were mixed in the absence of allosteric effectors, no complex was visible. After addition of ATP, a new complex of ~200 kDa appeared (*Figure 5c*). To verify its composition and stoichiometry, we analyzed fractions eluted from SEC on SDS PAGE. Both NrdA and NrdB were visible on the gel in a 1:1 molar ratio (*Figure 5c* insert). This complex, formed in conditions promoting active RNR, is conceivably an $\alpha_2\beta_2$ complex. After addition of the inhibiting effector dATP, a complex with a molecular mass consistent with $\alpha_4\beta_4$ eluted. Typical for complexes that equilibrate between each other faster than the time of analysis, a titration using increasing concentrations of NrdA-NrdB mixtures in the presence of 100 μM dATP showed a gradual movement of the NrdB tetramer peak to the left up to the position of an $\alpha_4\beta_4$ complex rather than showing distinct $\beta_4$, $\alpha_2\beta_4$ and $\alpha_4\beta_4$ peaks (*Figure 5—figure supplement 1d*).

## Three-dimensional structure of *L. blandensis* NrdB

The crystal structure of the dATP-inhibited complex of *L. blandensis* NrdB at 2.45 Å resolution (PDB 5OLK) revealed a novel tetrameric arrangement hitherto not observed in the RNR family, with approximate 222 point symmetry (*Figure 6a*, *Table 1*). Each monomer consists of an ATP-cone domain (residues 1–103) joined by a short linker (104-106) to a metal-binding α-helical core domain (residues 107–398) and a disordered C-terminus (399–427). The latter two features are typical of the NrdB/F family. This domain arrangement gives the NrdB monomer and dimer extended conformations that are presumably flexible in solution (*Figure 6b*). The dimer buries about 1100 Å$^2$ of solvent accessible area on each monomer. The tetrameric arrangement is completely dependent on interactions between the ATP cone domains, as no contacts are made between the core domains in the two dimers that make up the tetramer (*Figure 6a*).

The ATP-cone domain in *L. blandensis* NrdB is structurally very similar to the one recently identified in the NrdA protein of *P. aeruginosa* (*Johansson et al., 2016*). The root-mean-square deviation for 92 equivalent Cα atoms is 1.2 Å. The electron density unambiguously confirms the *L. blandensis* ATP-cone's ability to bind two molecules of dATP, which it shares with *P. aeruginosa* NrdA (*Figure 6c*). Despite a local sequence identity of only 31% to *P. aeruginosa* NrdA, all amino acids involved in binding both dATP molecules are conserved (*Figure 6c*). The two dATP molecules bind in a 'tail-to-tail' arrangement that orients the base of the 'non-canonical' dATP towards the fourth, most C-terminal helix, an arrangement made possible by the binding of a Mg$^{2+}$ ion between the triphosphate moieties.

Remarkably, the interactions between the ATP-cones in *L. blandensis* NrdB are also very similar to those seen in *P. aeruginosa* NrdA (*Johansson et al., 2016*), despite the fact that the ATP-cone is attached to a structurally completely different core domain. The main interactions occur between the last two helices α3 and α4 in respective ATP-cones (*Figure 6d*). A hydrophobic core in *L. blandensis* NrdB involving residues Met80, Ile92 and Ile93 in both monomers is reinforced by salt bridges between residues Asp73, Asp76 in one monomer and Lys89 in the other. The two domains bury ~510 Å$^2$ of solvent accessible area. This is slightly less than the ~640 Å$^2$ buried in the equivalent interaction involving the ATP-cones of *P. aeruginosa* NrdA. Within each ATP cone, helices α3 and α4 have the same relative orientation, partly determined by an internal salt bridge between the conserved Asp73 and Arg95, but the helix pair in *L. blandensis* NrdB is rotated relative to its

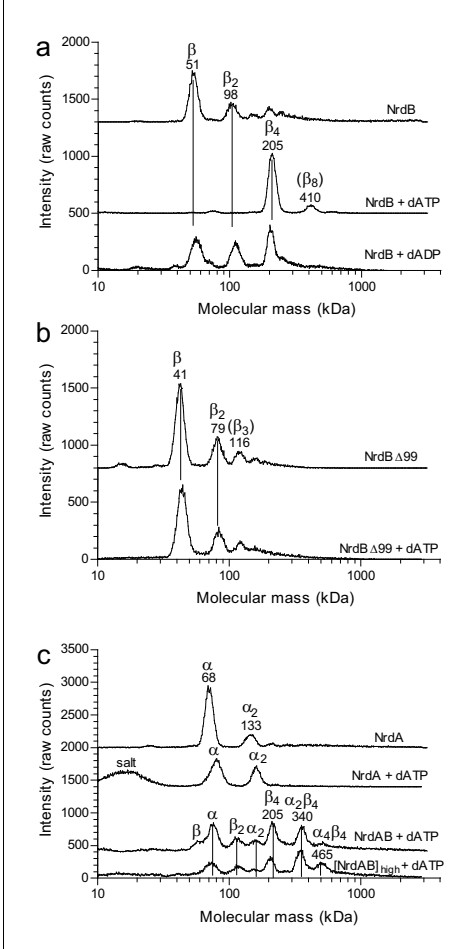

**Figure 4.** GEMMA analysis of the *L. blandensis* RNR subunits α, β and their combinations in the presence and the absence of allosteric effectors. (**a**) 0.1 mg/ml NrdB (~2 μM) analyzed in the absence and presence of 50 μM dATP or dADP. (**b**) The NrdBΔ99 mutant analyzed in the absence and the presence of 50 μM dATP. (**c**) In the top two traces 0.1 mg/ml NrdA (~1.4 μM) is analyzed in the absence or presence of 100 μM dATP. In the third trace, 0.1 mg/ml NrdB is added to the mixture. The last trace is similar to the third trace, but the concentration of NrdA and NrdB is increased to 0.4 mg/ml NrdA and 0.3 mg/ml NrdB (~6 μM of each). The analyses of NrdA-NrdB complexes were performed at a very low pressure (1.4 Psi) to avoid the influence of magnesium-nucleotide clusters on the measurement and to minimize the risk of false protein-protein interactions. The baselines of the individual experiments are distributed in the vertical direction to be able to fit many traces in each panel.

DOI: https://doi.org/10.7554/eLife.31529.006

The following figure supplement is available for figure 4:

**Figure supplement 1.** GEMMA measurements showing the dependence of NrdB tetramerization on the dATP concentration.

DOI: https://doi.org/10.7554/eLife.31529.007

counterpart in the other ATP cone by about 15° compared to *P. aeruginosa* NrdA, which reduces the number of possible interactions between them. Interestingly, the dATP-induced tetramer leaves free the surfaces of both dimers of *L. blandensis* NrdB that are thought to interact with the NrdA subunit in productive RNR complexes, which implies that one or two dimers of *L. blandensis* NrdA could attach to these surfaces in a near-productive fashion in $\alpha_2\beta_4$ or $\alpha_4\beta_4$ complexes (*Figure 6a*).

Two metal ions are found to bind to each of the monomers of *L. blandensis* NrdB. Comparison of their B-factors with those of surrounding atoms suggest that they are fourth row elements with close to full occupancy, but does not enable us to distinguish between Mn, Fe or Ca. No metal ions were added to the protein preparation, but crystals were obtained in 0.2 M $CaCl_2$. The distance between metal ions varies between 3.4–3.8 Å in the four monomers (*Figure 6—figure supplement 1*), but the metal coordination is very similar. The coordination distances are long for an RNR metal center, with the shortest distances being 2.3–2.4 Å, in contrast to the more typical 1.9–2.1 Å seen in other NrdB proteins containing Mn or Fe. Anomalous diffraction at three wavelengths was used to determine the nature of the metal. *Figure 6—source data 1* shows that there is little difference in the peak heights at the metal ion positions in anomalous difference maps irrespective of wavelength, which strongly suggests that the metal site is occupied by $Ca^{2+}$ from the crystallization medium. Furthermore, the peaks were only about twice the height of those from well-ordered sulfur atoms in the structure. At these wavelengths, S has an anomalous f' component of about 0.5 electrons. For Mn or Fe one would expect an anomalous signal around eight times that of S (*Figure 6—source data 2*). The presence of $Ca^{2+}$ is also consistent with the long coordination distances. Furthermore, an X-ray fluorescence spectrum of the crystal (not shown) revealed only traces of Mn or Fe.

Interestingly, Tyr207 is found near the metal site at the position expected for a radical-carrying Tyr, but it is not hydrogen-bonded to the metal site, its hydroxyl group being at around 6 Å from the side chain of Glu170. Tyr207 is H-bonded to the side chain of Thr294, but the latter is not H-bonded to a metal center ligand. On the other side of the metal site, Trp177 is H-bonded to the side chain of Glu263. This tryptophan is completely conserved in the NrdBi subclass (*Figure 6—figure supplement 2*). It makes the same interaction as Trp111 from *E. coli* NrdB, despite

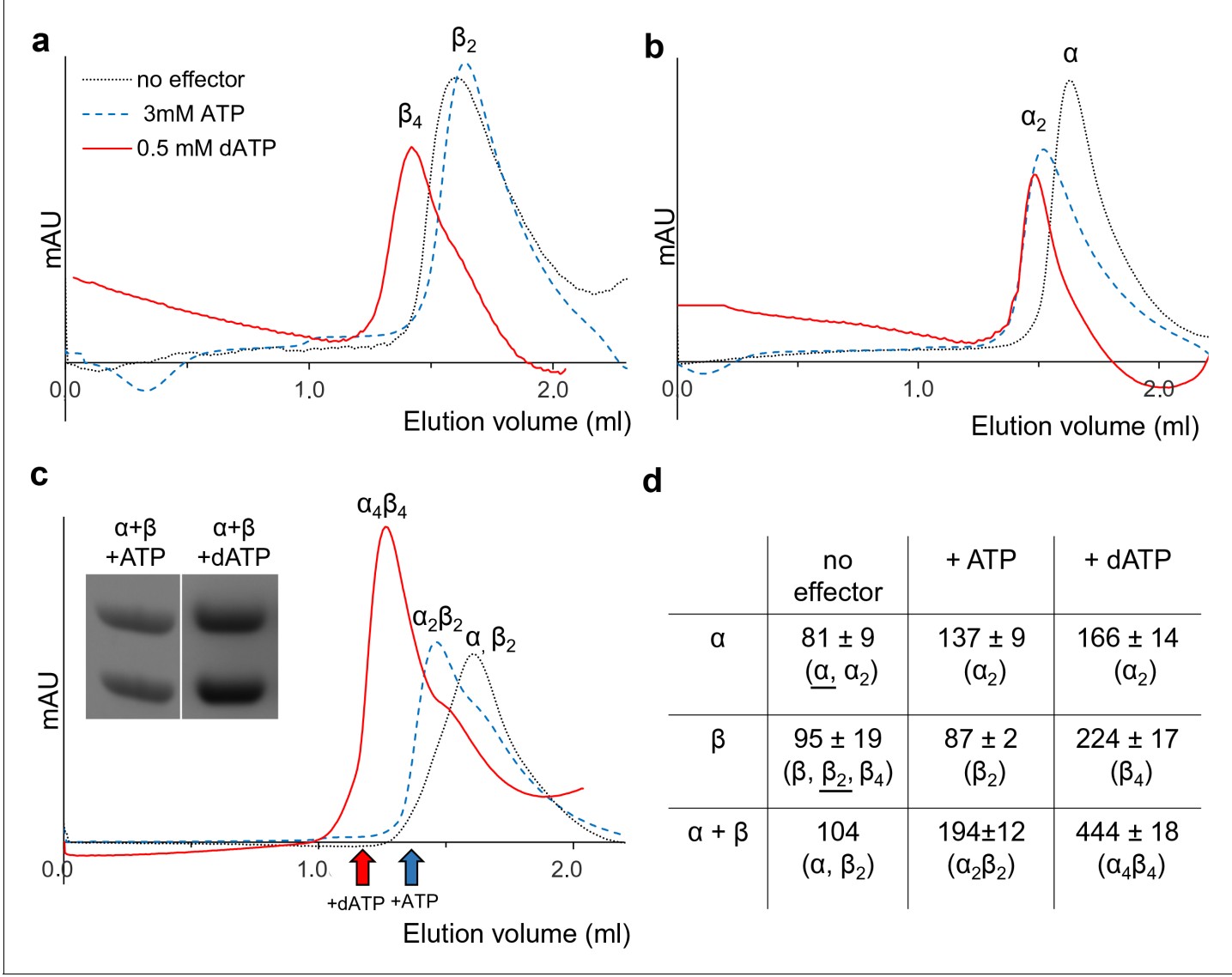

**Figure 5.** Size exclusion chromatography of *L. blandensis* RNR in the absence and presence of 3 mM ATP or 0.5 mM dATP. Proteins at concentrations of 20 μM each were pre-incubated separately or mixed together for 10 min at room temperature in the absence (black dotted line) or presence of ATP (blue dashed line) or dATP (red solid line), centrifuged and applied to the column equilibrated with SEC buffer at 7°C. Panels show NrdB (**a**), NrdA (**b**), and an equimolar mixture of NrdA and NrdB (**c**). C Insert: SDS PAGE of the eluted proteins run in the presence and absence of the indicated effectors. Elution positions of fractions applied to gel are indicated by *red* (in the presence of dATP) and *blue* (in the presence of ATP) arrows, respectively. (**d**) Summary of multiple SEC experiments varying protein concentrations of NrdA (10–113 μM), NrdB (5–150 μM) and mixtures of NrdA and NrdB at ratios of 1:1 or 1:2. Molecular masses and standard deviations are calculated for 3–5 experiments, and closest estimated complex stoichiometry are shown in parenthesis, with major species underlined when appropriate (see *Figure 5—figure supplement 1* for details).
DOI: https://doi.org/10.7554/eLife.31529.008

The following figure supplement is available for figure 5:

**Figure supplement 1.** Size exclusion chromatography studies of the *L. blandensis* NrdA and NrdB at low protein concentration.
DOI: https://doi.org/10.7554/eLife.31529.009

the fact that it is projected from the first helix of the 4-helix bundle containing the metal center ligands, rather than the second helix. The first helix of the bundle has a very unusual distortion in the middle (*Figure 6—figure supplement 3*). Normally this is an undistorted α-helix, but in *L. blandensis* NrdB, residues 171–175 form a loop that bulges away from the metal site, with the exception of Lys174, whose side chain is projected towards the metal site and is H-bonded to Glu170 (*Figure 6e*). The significance of this distortion is at present not clear. However we are certain that

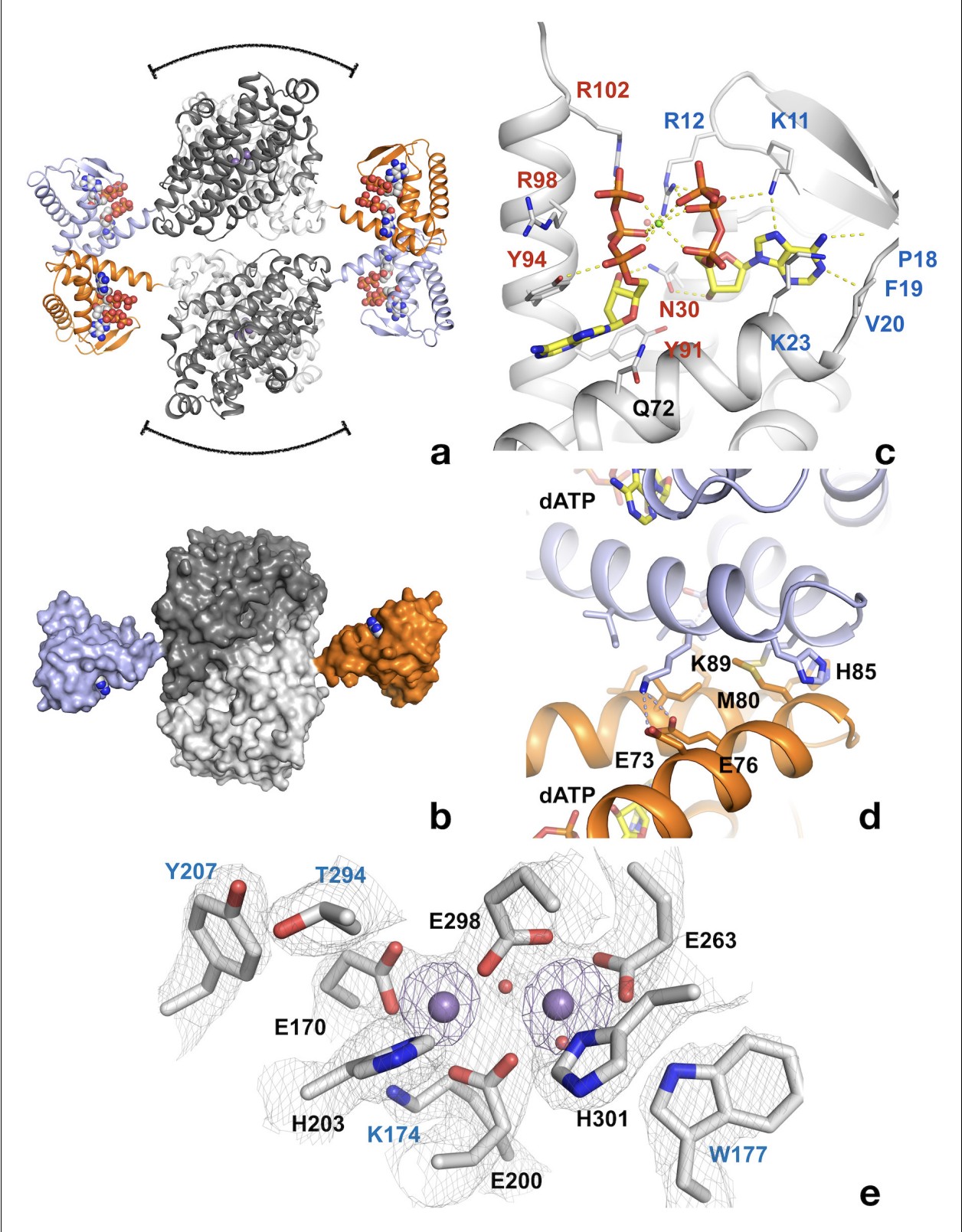

**Figure 6.** Structure of tetrameric *L. blandensis* NrdB in complex with dATP. (a) Overall structure of the NrdB tetramer. The two monomers of each dimer are colored in different shades of gray, while the N-terminal ATP-cones are colored orange and light blue respectively. The dATP molecules are shown in CPK representation. The curved lines at the top and bottom of each dimer core indicate the proposed binding area in the active $\alpha_2\beta_2$ complex of RNRs. (b) Surface representation of an NrdB dimer extracted from the tetramer structure, showing the presumably flexible nature of the

*Figure 6 continued on next page*

*Figure 6 continued*

ATP-cones in solution. The color scheme is as in panel a). (c) Binding of two dATP molecules to the ATP-cone. The dATP molecules are shown as sticks. Residues involved in binding the two dATP molecules are labeled in blue and red respectively. Polar contacts are shown as dotted yellow lines. (d) The interface between ATP-cones in the NrdB tetramer. Chains A and D are shown in light blue and orange respectively. Side chains of residues involved in the interface are labeled. The two dATP molecules closest to the interface are shown as sticks. Polar contacts are shown as yellow dotted lines. (e) Structure of the metal site in chain C, which is representative of the others. Metal ions are shown as purple spheres. 2 m|Fo|-D|Fc| electron density is shown as a grey mesh, contoured at 1.4 σ. Asn m|Fo|-D|Fc| OMIT map for the metal ions is shown in purple, contoured at 4.0 σ.
DOI: https://doi.org/10.7554/eLife.31529.010

The following source data and figure supplements are available for figure 6:

**Source data 1.** Data quality statistics for the anomalous data used for metal identification.
DOI: https://doi.org/10.7554/eLife.31529.014

**Source data 2.** Peak heights of metal ions in anomalous difference maps calculated at three different wavelengths after refining the same structure against data at each wavelength.
DOI: https://doi.org/10.7554/eLife.31529.015

**Figure supplement 1.** The ligands of the metal center in the four non-crystallographically related monomers of *L. blandensis* NrdB.
DOI: https://doi.org/10.7554/eLife.31529.011

**Figure supplement 2.** NrdBi HMMER sequence profile logo.
DOI: https://doi.org/10.7554/eLife.31529.012

**Figure supplement 3.** Distortion of the first helix of the metal-coordinating 4-helix bundle in *L. blandensis* NrdB.
DOI: https://doi.org/10.7554/eLife.31529.013

the distortion is not caused by the coordination of $Ca^{2+}$ instead of the native metal, as the structure is the same in crystals of the N-terminally truncated form grown in the presence of $Mn^{2+}$, which will be presented elsewhere.

The nature of the native metallo-cofactor was addressed by X-band EPR spectroscopy and catalytically active samples were analyzed at 5–32 K. The spectra revealed a mixture of signals from low and high-valent manganese species (*Figure 7a*). In particular, at 5 K a 6-line signal attributable to low valent Mn ($Mn^{II}$) was clearly visible, overlaid with a complex multiline signal with a width of approximately 1250G. Increasing the temperature to 32 K resulted in a significant decrease of the latter signal, while the 6-line feature remained relatively intense (*Figure 7a*, top and middle). The shape, width and temperature dependence of the multiline signal are all highly reminiscent of the signal reported for super-oxidized manganese catalase as well as the 16-line signal observed during the assembly of the dimanganese/tyrosyl radical cofactor in NrdF RNR, and are attributable to a strongly coupled $Mn^{III}Mn^{IV}$ dimer ($S_{Total} = \frac{1}{2}$) (*Figure 7a*, bottom) (*Cotruvo et al., 2013*; *Zheng et al., 1994*). The low valent species appeared to be weakly bound to the protein, while the $Mn^{III}Mn^{IV}$ dimer was retained in the protein following desalting (*Figure 7—figure supplement 1*). The observation of a high-valent Mn cofactor is consistent with the Mn-dependent increase in catalytic activity of the enzyme and its inhibition by the addition of $Fe^{2+}$ (*Figure 7b–c*), and is suggestive of a novel high-valent homodimeric Mn-cofactor. Indeed, earlier calculations have shown that high-valent Mn-dimers have reduction potentials similar to that of the tyrosyl radical in standard class I RNRs, but are hitherto not observed in RNRs (*Roos and Siegbahn, 2011*). A more detailed biophysical characterization of this cofactor is currently ongoing.

## *L. blandensis* NrdB binds two nucleotide molecules per ATP-cone

The finding that two dATP molecules were bound to the ATP-cone in the 3D structure prompted us to investigate nucleotide binding using isothermal titration calorimetry (ITC). Binding curves for dATP, ATP and dADP to NrdB, including a reverse titration of NrdB to dATP, were all consistent with a single set of binding sites except for the ATP-cone deletion mutant (NrdBΔ99), which did not bind nucleotides at all (*Figure 8*). In all other titrations the fitted apparent N value was significantly above one, but to be meaningful the N value needs to be an integer. Using information available from the 3D structure, a fixed stoichiometry of 2 was used to fit dATP binding to NrdB. The fit indicated 59% active (i.e. nucleotide binding) protein. A fixed stoichiometry of 1 would result in higher than 100% active protein concentration, which is impossible. Fitting dADP and ATP with a 59% proportion of active protein taken from the dATP experiment, we could calculate stoichiometries of 2.2 and 2.1 for ATP and dADP respectively, indicating that each NrdB monomer can bind two molecules

**Table 1.** Data collection and refinement statistics.
Figures in parentheses are for the highest resolution shell.

| Wavelength | 0.9724 |
| --- | --- |
| Resolution range (Å) | 44.7–2.45 (2.51–2.45) |
| Space group | P1 |
| Unit cell (Å, °) | a = 74.0, b = 90.5, c = 90.9, α = 110.78, β = 98.99, γ = 114.11 |
| Total no. reflections | 246972 (15753) |
| No. unique reflections | 67728 (4517) |
| Multiplicity | 3.6 (3.5) |
| Completeness (%) | 97.8 (97.3) |
| Mean I/σ(I) | 7.4 (1.0) |
| Wilson B-factor (Å$^2$) | 63.3 |
| $R_{merge}$ (I) | 0.093 (1.058) |
| $R_{meas}$ (I) | 0.109 (1.243) |
| $R_{pim}$ (I) | 0.055 (0.639) |
| CC(½) (I) | 0.997 (0.495) |
| Reflections used in refinement | 67723 |
| Reflections used for $R_{free}$ | 3353 (5%) |
| $R_{model}$ (F) | 0.205 (0.221) |
| $R_{free}$ (F) | 0.238 (0.265) |
| Number of non-hydrogen atoms: | |
| macromolecules | 12919 |
| ligands | 284 |
| water molecules | 129 |
| rms deviation from ideal geometry (bonds) | 0.010 |
| rms deviation (angles) | 1.11 |
| Ramachandran favored (%) | 97.7 |
| Ramachandran allowed (%) | 2.1 |
| Ramachandran outliers (%) | 0.2 |
| Rotamer outliers (%) | 3.5 |
| Clashscore | 3.22 (100th percentile) |
| Wilson B-factor (Å$^2$) | 63.3 |
| Average B-factors (Å$^2$): | |
| macromolecules | 66.5 |
| ligands | 69.5 |
| solvent | 54.0 |

DOI: https://doi.org/10.7554/eLife.31529.016

of adenosine nucleotides (*Figure 8f*). $K_d$ for the three different nucleotides (*Figure 8f*) indicated a 38-fold and 15-fold lower affinity for ATP and dADP compared to dATP. Thermodynamic parameters (*Figure 8f*) indicated that the interactions are enthalpy-driven, with negative ΔH values of −79,−44 and −103 for dATP, ATP and dADP respectively.

## Discussion

Although allosteric regulation is built into many enzymes (*Gunasekaran et al., 2004*), it can evolve in a surprisingly dynamic way (*Aravind and Koonin, 1999*; *Lang et al., 2014*; *Lundin et al., 2015*). This has also been shown experimentally by grafting an allosteric domain to an enzyme that was

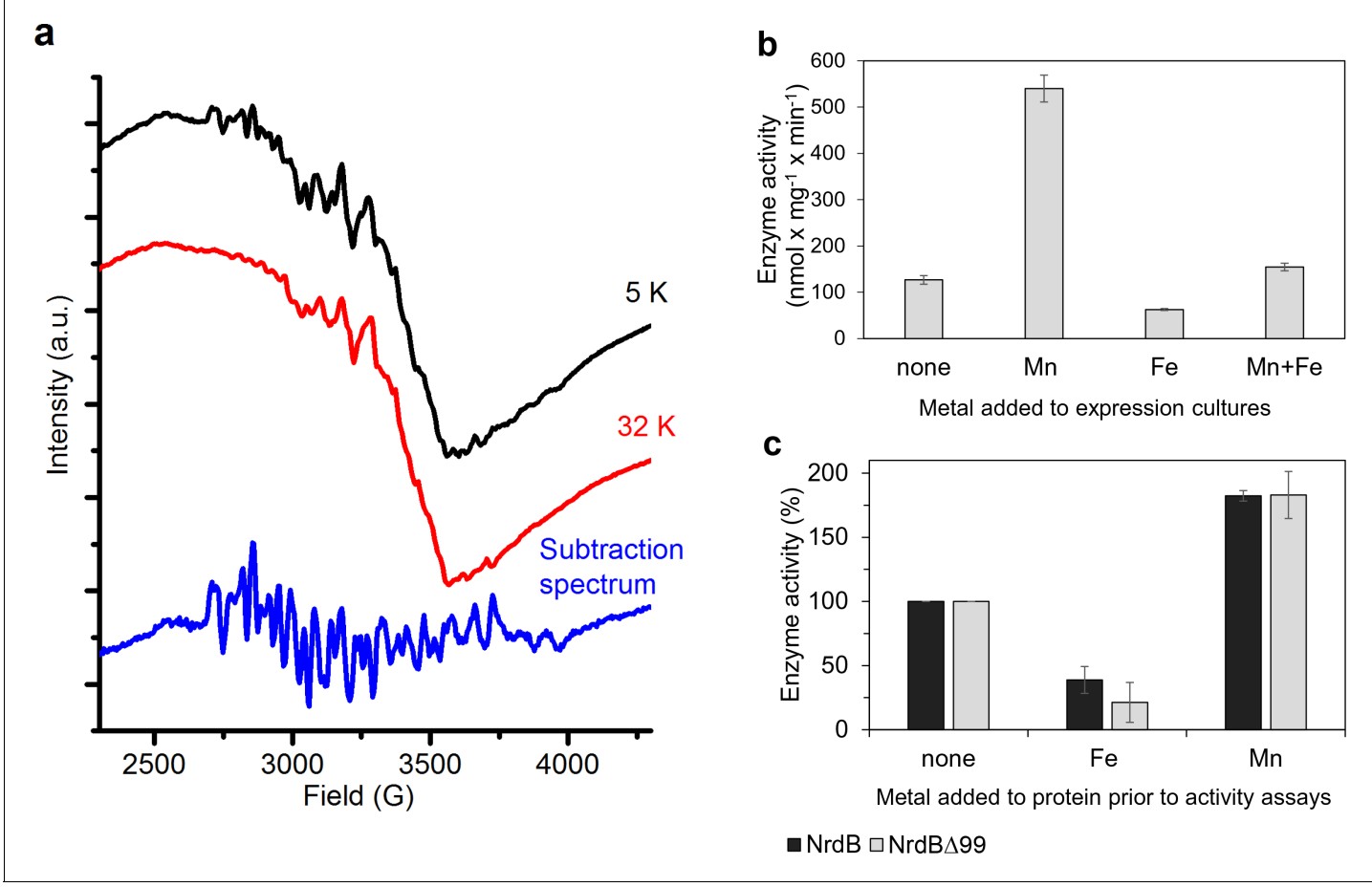

**Figure 7.** Type of dinuclear metal center of *L. blandensis* NrdB and metal-dependency of enzyme activity. (**a**) X-band EPR spectra of catalytically active, non-reconstituted, samples recorded at 5 K (*black*, top); 32 K (*red*, middle) (signal intensity multiplied by 3.7 for clarity; multiline spectrum obtained by subtraction of a scaled 32K spectrum from the 5K spectrum (*blue*, bottom) (signal intensity multiplied by three for clarity). Instrument settings: microwave frequency = 9.28 GHz; power = 1 mW; modulation amplitude = 10G; modulation frequency = 100 kHz. (**b**) Enzyme activity of NrdBΔ99 purified from heterologously expressed cultures grown with addition of different divalent metal ions as indicated; the Mn-sample was used for the EPR analysis. (**c**) Enzyme activity was measured after addition of a total concentration of 20 µM divalent metal ions to 10 µM of wild-type or NrdBΔ99 protein as indicated. Enzyme activity without addition of metals was set as 100% and corresponded to 592 and 217 nmol mg$^{-1}$ min$^{-1}$ for wild type and NrdBΔ99 enzymes respectively. Error bars indicate the standard deviation of three measurements.

DOI: https://doi.org/10.7554/eLife.31529.017

The following figure supplement is available for figure 7:

**Figure supplement 1.** X-band EPR spectra recorded at 5 K and 32 K of catalytically active samples before and after desalting treatment.

DOI: https://doi.org/10.7554/eLife.31529.018

previously not allosterically regulated (*Cross et al., 2013*). The presence of an ATP-cone in the radical-generating subunit of *L. blandensis* RNR, provides, to our knowledge, the first example of another degree of evolutionary dynamics by the transfer of a domain conferring allosteric regulation to a non-homologous component of an enzyme complex.

It is critically important for an organism to control the supply of dNTPs to allow fidelity in DNA replication and repair (*Mathews, 2006*). Specificity regulation of RNR makes sure relative concentrations of dNTPs fit the organism's DNA composition. On the other hand, activity regulation assures that absolute concentrations of dNTPs follow the different requirements through the cell cycle (*Hofer et al., 2012*). Specificity regulation of RNRs is ubiquitous, integrated in the catalytic subunit of the enzyme, and works via the classical homooligomeric model of allosteric regulation (*Andersson et al., 2000*; *Hofer et al., 2012*; *Larsson et al., 2004*; *Reichard, 2010*; *Swain and Gierasch, 2006*; *Torrents et al., 2000*; *Zimanyi et al., 2016*). In contrast, the activity regulation is

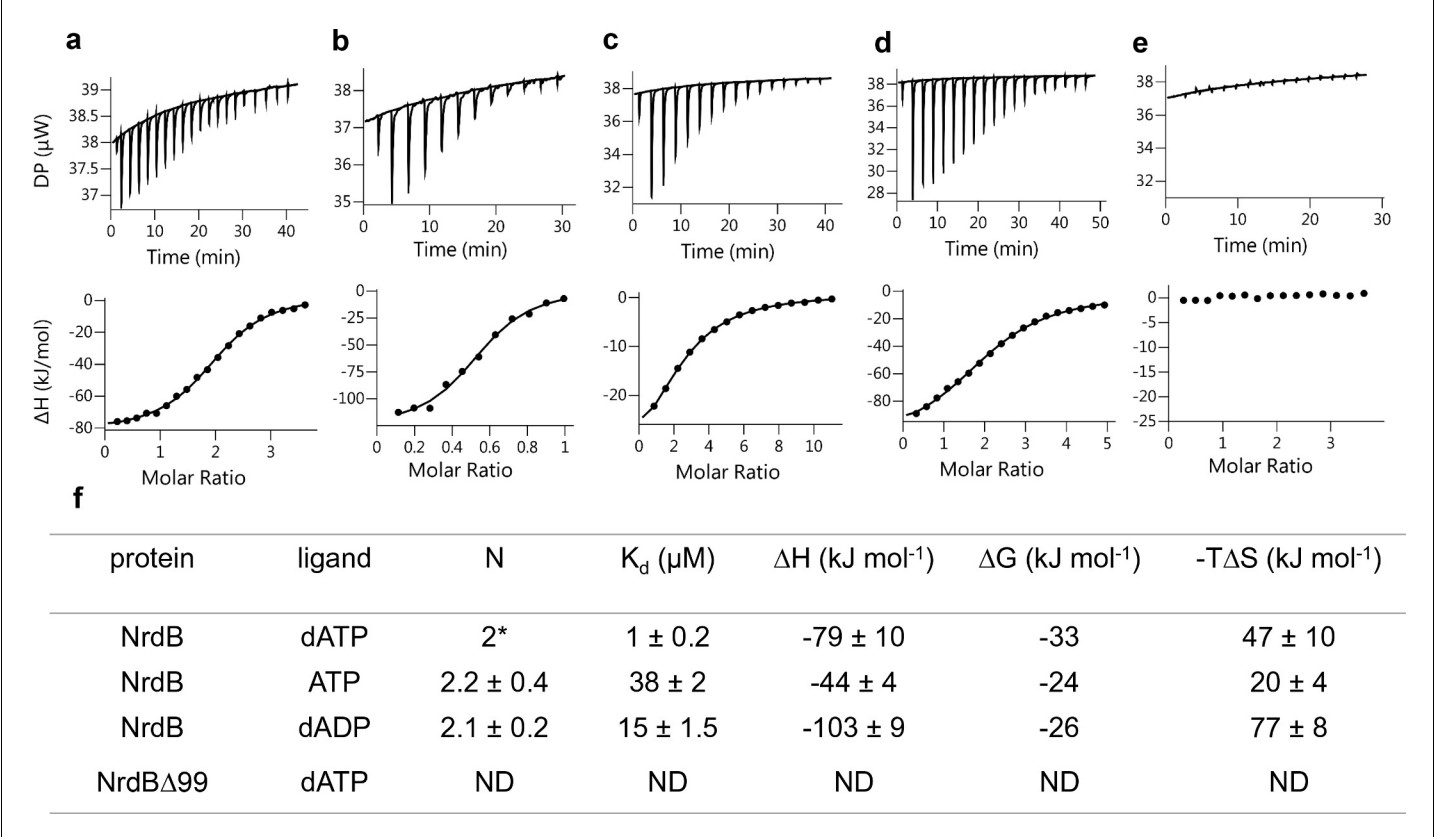

**Figure 8.** Representative ITC thermograms obtained by titration of dATP (**a**), ATP (**c**) and dADP (**d**) into NrdB. Isothermal calorimetric enthalpy changes (upper panel) and resulting binding isotherms (lower panel) are shown. Reverse titration of NrdB to dATP (**b**). Titration of dATP to NrdBΔ99 (**e**). Thermodynamic parameters of ligand binding to NrdB (**f**). Binding isotherms were fitted using a one-set-of sites binding model. Values are reported as the mean ±SD of three titrations (and additional two reverse titrations for dATP). All titrations were performed at 10°C as described in Materials and methods. *=binding stoichiometry was kept constant N = 2, considering monomeric protein concentrations. ND = not detected.
DOI: https://doi.org/10.7554/eLife.31529.019

controlled by an accessory domain, the ATP-cone, and works by affecting the distribution of the holoenzyme heteromeric complexes. Moreover, the ATP-cone is only found in some RNRs, and appears to be gained by domain shuffling when evolutionary selection favours it and lost when selection decreases (*Lundin et al., 2015*). These dynamics are further evidenced by the differences in mechanisms recently discovered in class I RNRs (*Ando et al., 2011*, *2016*; *Fairman et al., 2011*; *Johansson et al., 2016*; *Jonna et al., 2015*). The current study was prompted by the interesting observation that several radical-generating subunits of the NrdBi subclass possess an N-terminal ATP-cone and that a few radical-generating subunits of the NrdF subclass possess a C-terminal ATP-cone. Our discovery evokes several pertinent questions: does the ATP-cone fused to a radical-generating subunit function as an allosteric on/off switch; how does it affect the distribution of holoenzyme complexes under active and inhibited conditions; is its structural mode of action similar to that of ATP-cones fused to the catalytic subunit?

We have delineated the function of the ATP-cone that is N-terminally fused to the *L. blandensis* NrdBi protein. In the presence of the positive effector ATP, *L. blandensis* NrdBi was a dimer, which by interaction with the *L. blandensis* catalytic subunit NrdAi formed the common active $\alpha_2\beta_2$ complex, also found in e.g. *E. coli*, *P. aeruginosa* and eukaryotic class I RNRs. Binding of dATP to the ATP-cone instead promoted oligomerization of *L. blandensis* NrdBi to an inactive $\beta_4$ complex, with a novel tetrameric structure revealed by crystallography. This oligomerization is reminiscent of the 'ring-shaped' $\alpha_4$ and $\alpha_6$ complexes formed by dATP binding to the NrdA-linked ATP-cones in *P. aeruginosa* and eukaryotic RNRs. When *L. blandensis* NrdAi was added to the dATP-loaded NrdBi tetramer, higher molecular mass complexes of $\alpha_2\beta_4$ and $\alpha_4\beta_4$ appeared. The NrdA dimers seem to bind

to the NrdB tetramers in a 'nonproductive' orientation, and in other studied RNRs it usually means that the structure of the complexes does not allow efficient electron transfer between NrdA and NrdB. The crystal structure shows that the tetramerization of *L. blandensis* NrdBi leaves the putative interaction surface for NrdAi free, which is consistent with the possibility to form both $\alpha_2\beta_4$ and $\alpha_4\beta_4$ oligomers. However the structure does not suggest the structural basis for a disruption of the cysteinyl radical generation pathway in these non-productive complexes.

The structure of the dATP-loaded *L. blandensis* NrdB shows that it binds two dATP molecules per ATP-cone. Both molecules bind to the same site and interact with each other through a $Mg^{2+}$ ion. Most allosterically regulated RNRs characterized so far bind only one dATP molecule per ATP-cone. However, a novel class of ATP-cones that binds two dATP molecules was recently characterized in *P. aeruginosa* NrdA (*Johansson et al., 2016*; *Jonna et al., 2015*). The NrdB-linked ATP-cone of *L. blandensis* has sequence motifs characteristic of this kind of ATP-cone. The structure confirms that both dATP molecules bind essentially identically as in *P. aeruginosa* NrdA and that the ATP-cones make similar interactions to each other, distinct from those made in the eukaryotic $\alpha_6$ complexes. It has also been shown in the RNR transcriptional regulator NrdR, that ATP and dATP bind with positive cooperativity to its ATP-cone (*McKethan and Spiro, 2013*), implying binding of more than one nucleotide molecule.

The ATP-cone in *L. blandensis* NrdBi offers a unique possibility to measure its binding capacity for (deoxy)adenosine nucleotides. This has not been possible in earlier studied RNRs, where the ATP-cone is bound to the α subunit that also possesses a binding site for allosteric regulation of substrate specificity. ITC ligand binding studies confirmed that the *L. blandensis* ATP-cone bound two dATP molecules, and showed that it also can bind two molecules of ATP or two molecules of dADP. Structurally, binding of dADP is not surprising, since the γ-phosphates of the dATP molecules make only one interaction with the protein, through Arg102, and they only contribute 2 of the 6 coordinating atoms of the intervening $Mg^{2+}$ ion. Nonetheless, this is the first observation of dADP binding to the ATP-cone of an RNR enzyme. dADP inhibits enzyme activity in a similar manner to dATP, but higher concentrations are required. In vivo, deoxyribonucleoside diphosphates are rapidly converted to triphosphates and cellular dADP concentrations are very low (*Mathews, 2014*; *Traut, 1994*). Nevertheless, dADP is one of the products of *L. blandensis* RNR, and perhaps local concentrations are higher. The ability of dADP to regulate the activity of *L. blandensis* RNR may enable it to react more rapidly to changes in (deoxy)adenosine nucleotide concentrations and provide the cell with a fitness advantage. It can therefore not be excluded that dADP has a physiological role, although we think that the 15 times stronger affinity for dATP suggests that it is the major inhibitory nucleotide effector. Binding of deoxyadenosine di- and monophosphates has been described for the ATP-cone in NrdR (*Grinberg et al., 2006*; *McKethan and Spiro, 2013*).

Based on the variable occurrence of the ATP-cone in RNR catalytic subunits, we have earlier hypothesized that its presence in RNRs is part of a dynamic process of gains and losses on a relatively short evolutionary time scale (*Lundin et al., 2015*). This suggests that the ATP-cone, contrary to what might be expected for a domain involved in allostery, does not require a long evolutionary period of integration with a protein to contribute to regulation and that it hence lends itself to a highly dynamic evolution of allosteric activity control (*Lundin et al., 2015*). This hypothesis is nicely supported by the N-terminally positioned ATP-cone, in the radical-generating subunit, NrdB, of *L. blandensis* RNR described here. No RNR in the phylogenetic subclass NrdAi/Bi contains an ATP-cone in the catalytic subunit, and only a minority – mostly encoded by Flavobacteriia, a marine class of Bacteroidetes – have an NrdB with an ATP-cone, suggesting the relatively recent acquisition of the ATP-cone to an RNR subclass that is otherwise not activity regulated (*Figure 1*). Moreover, in the Ib subclass, which also lacks ATP-cones in the catalytic subunit, we detected a C-terminally positioned ATP-cone in the radical-generating subunit. This was found in only two sequences from closely related organisms and might hence be an example of an even more recent evolutionary event.

The evidence we have presented here for an ATP/dATP-sensing master switch of the *L. blandensis* NrdAi/NrdBi class I RNR, suggests a potential for the use of ATP-cones to control the activity of engineered proteins. Multimeric enzymes could be inactivated through sequestration of one member of an active complex, by control of dATP concentrations in the reaction mixture.

The surprises did not end with the discovery of a functional master switch in the *L. blandensis* NrdAi/NrdBi RNR. The active center of the radical generating subunit of class I RNRs have earlier

been found to consist either of an $Fe^{III}Fe^{III}$ or $Mn^{III}Mn^{III}$ pair coupled to a tyrosine residue acting as long-term storage for the catalytically essential radical (*Berggren et al., 2017*; *Cotruvo and Stubbe, 2012*), or a $Fe^{III}Mn^{IV}$ center not coupled to an amino acid radical (*Griese et al., 2014*). Although further analyses are required to fully characterize the *L. blandensis* NrdBi active center, it appears to present a fourth type, a high-valent $Mn^{III}Mn^{IV}$ center that lacks a suitably positioned radical storage amino acid. The evolutionary flexibility displayed by the ATP-cone appears hence all but equaled by the evolutionary tuning possibilities of the metal centers in the radical generating subunit of class I RNR.

## Materials and methods

### Key resources table

| Reagent type (species) or resource | Designation | Source or reference | Identifiers |
|---|---|---|---|
| Gene (*Leeuwenhoekiella blandensis* sp. nov. MED 217) | NrdAi | NA | WP_009781766 |
| Gene (*Leeuwenhoekiella blandensis* sp. nov. MED 217) | NrdBi | NA | EAQ51288 |
| Strain, strain background (*Leeuwenhoekiella blandensis*) | *Leeuwenhoekiella blandensis* sp. nov. MED 217 | doi: 10.1099/ijs.0.64232–0 | |
| Strain, strain background (*Escherichia coli*) | *Escherichia coli* BL21(DE3) | (Novagen) Merck | |
| Strain, strain background (*Escherichia coli*) | *Escherichia coli* DH5α | ThermoFisher Scientific | |
| Recombinant DNA reagent | pET-28a(+) vector (Novagen) | Merck | |
| Software, algorithm | HMMER | doi:10.1371/journal.pcbi.1002195 | RRID:SCR_005305 |
| Software, algorithm | RefSeq | | RRID:SCR_003496 |
| Software, algorithm | ProbCons | doi:10.1101/gr.2821705 | RRID:SCR_011813 |
| Software, algorithm | FastTree | doi:10.1371/journal.pone.0009490 | RRID:SCR_015501 |
| Software, algorithm | XDS | doi:10.1107/s0907444909047337 | RRID:SCR_015652 |
| Software, algorithm | CCP4 | doi:10.1107/s0907444910045749 | RRID:SCR_007255 |
| Software, algorithm | Phaser | doi:10.1107/s0021889807021206 | RRID:SCR_014219 |
| Software, algorithm | PDB | | RRID:SCR_012820 |
| Software, algorithm | Coot | doi:10.1107/s0907444910007493 | RRID:SCR_014222 |
| Software, algorithm | Buccaneer | doi:10.1107/s0907444906022116 | RRID:SCR_014221 |
| Software, algorithm | Refmac | doi:10.1107/s0907444996012255 | RRID:SCR_014225 |
| Software, algorithm | Buster | | RRID:SCR_015653 |
| Software, algorithm | MolProbity | doi:10.1107/s0907444909042073 | RRID:SCR_014226 |

### Cloning

DNA fragments encoding NrdAi (WP_009781766) and NrdBi (EAQ51288) were amplified by PCR from *Leeuwenhoekiella blandensis* sp. nov. strain MED217 genomic DNA, isolated as described previously (*Pinhassi et al., 2006*), using specific primers: NrdA: LBR1_For 5'- cgagCATATGAGAGAAAACACTACCAAAC-3' and LBR1_Rev 5'- gcaaGGATCCTTAAGCTTCACAGCTTACA-3'. NrdB: LBR2_For 5'-cgagCATATGAGTTCACAAGAGATCAAA-3', LBR2_REV 5'- gcaaGGATCCTTAAAATAAGTCGTCGCTG-3', The PCR products were purified, cleaved with NdeI and BamHI restriction enzymes and inserted into a pET-28a(+) expression vector (Novagen, Madison, Wisconsin, USA). The obtained constructs pET-*nrdA* and pET-*nrdB* contained an N-terminal hexahistidine (His) tag and thrombin cleavage site. To construct a truncated NrdB mutant, lacking the entire ATP-cone domain, new forward primer LBR2Δ99_For 5'-cgatCATATGCTGGAGCGTAAAACAAAT-3' was used with LBR2_REV to yield a pET-*nrdB*Δ99. The cloning process and the resulting construct was similar to

that of the wild type protein, except that it lacked sequence coding for the N-terminal 99 amino acids.

## Protein expression

Overnight cultures of *E. coli* BL21(DE3)/pET28a(+) bearing pET-*nrdA*, pET-*nrdB* and pET-*nrdBΔ99* were diluted to an absorbance at 600 nm of 0.1 in LB (Luria-Bertani) liquid medium, containing kanamycin (50 µg/ml) and shaken vigorously at 37°C. At an absorbance at 600 nm of 0.8, isopropyl-β-D-thiogalactopyranoside (Sigma) was added to a final concentration of 0.01 mM for NrdA expression and 0.5 mM for NrdB and NrdBΔ99 expression. For particular experiments, 0.5 mM MnSO$_4$ or 0.5 mM FeNH$_4$(SO4)$_2$ or the combination of both metals (0.4 mM and 0.25 mM respectively) were added to NrdBΔ99 cultures. The cells were grown overnight at 14°C for NrdA expression and 20°C for NrdB and NrdBΔ99 expression and harvested by centrifugation.

## Protein purification

The cell pellet was resuspended in lysis buffer: 50 mM Tris-HCl pH 7.6 containing 300 mM NaCl, 20% glycerol, 10 mM imidazole, 1 mM PMSF. Cells were disrupted by high pressure homogenization and the lysate was centrifuged at 18,000 × g for 45 min at 4°C. The recombinant His-tagged protein was first isolated by metal-chelate affinity chromatography using ÄKTA prime system (GE Healthcare): the supernatant was loaded on a HisTrap FF Ni Sepharose column (GE Healthcare), equilibrated with lysis buffer (w/o PMSF), washed thoroughly with buffer and eluted with buffer containing 500 mM imidazole. Further purification was accomplished by fast protein liquid chromatography (FPLC) on a 125 ml column packed with Superose 12 Prep Grade or HiLoad 16/600 Superdex 200 pg column (GE Healthcare) using ÄKTA prime system, equilibrated with buffer containing 50 mM Tris-HCl pH 7.6, 300 mM NaCl, 10–20% glycerol. Eluted protein was collected. In the case of NrdA, all purification steps were performed in the presence of 2 mM DTT. Protein concentration was determined by measuring the UV absorbance at 280 nm based on protein theoretical extinction coefficients 91,135, 46,870 and 39,420 M$^{-1}$ cm$^{-1}$ for NrdA, NrdB and NrdBΔ99 respectively. Proteins were concentrated using Amicon Ultra-15 centrifugal filter units (Millipore), frozen in liquid nitrogen and stored at −80°C until used.

For crystallization, NrdB was subsequently cleaved by thrombin (Novagen) to remove the hexahistidine tag. 41 mg NrdB was incubated with 25 U thrombin at 6°C for 3 hr, in a buffer containing 40 mM Tris-HCl pH 8.4, 150 mM NaCl, and 2.5 mM CaCl$_2$ in a total volume of 50 ml. Subsequently, imidazole to a final concentration of 20 mM was added and the reaction mixture was applied to a HisTrap FF Ni Sepharose column in a buffer containing 20 mM imidazole. Unbound protein was collected, concentrated and further purified by FPLC (see above) in a buffer containing Tris-HCl pH 7.6, 300 mM NaCl, and 10% glycerol. The thrombin-cleaved NrdB contained three additional residues (GlySerHis) that originated from the cleavage site of the enzyme at its N-terminus. Protein purity was evaluated by SDS–PAGE (12%) stained with Coomassie Brilliant Blue.

For EPR measurements, NrdBΔ99 was frozen in liquid nitrogen in EPR tubes directly after imidazole elution from the HisTrap column. Additional EPR samples were frozen after desalting the protein using PD-10 desalting columns (GE Healthcare).

## Enzyme activity assays

Enzyme assays were performed at room temperature in 50 mM Tris-HCl at pH 8 in volumes of 50 µl. Reaction conditions, giving maximal activity were determined experimentally. In a standard reaction the constituents were; 10 mM DTT, 10 mM MgAc, 10 mM KCl, 0.8 mM (or when indicated 3 mM) CDP, and various concentrations of allosteric effectors ATP, dATP or dADP. Mixtures of 0.5 µM NrdA and 2 µM wild type NrdB or NrdBΔ99 (for determination of s-site K$_L$) (*Figure 3a–b*) or 0.5 µM of NrdB and 2 uM NrdA (for determination of a-site K$_L$), (*Figure 3c–f*) were used. Some components were explicitly varied in specific experiments. Generally, 0.8 mM CDP was used as substrate. High CDP concentration (3 mM) was used for dADP titrations, to exclude potential product inhibition by binding of dADP to the active site. When dTTP was used as an s-site effector, 0.5 mM or 0.8 mM GDP was used as substrate. In four substrate assays, the four substrates CDP, ADP, GDP and UDP were simultaneously present in the mixture at concentrations of 0.5 mM each. The substrate mixture was added last to start the reactions. Certain assays were performed in the presence of Mn

(CH$_3$COO)$_2$ or FeNH$_4$(SO4): 20 µM of the indicated metal was added to 10 µM NrdB or NrdBΔ99 protein, incubated for 10 min, mixed with NrdA and added to the reaction mixture.

Enzyme reactions were incubated for 10–30 min and then stopped by the addition of methanol. The chosen incubation time gave a maximum substrate turnover of 30%. Substrate conversion was analyzed by HPLC using a Waters Symmetry C18 column (150 × 4.6 mm, 3.5 µm pore size) equilibrated with buffer A. 25 µl samples were injected and eluted at 1 ml/min with a linear gradient of 0–100% buffer B (buffer A: 10% methanol in 50 mM potassium phosphate buffer, pH 7.0, supplemented with 10 mM tributylammonium hydroxide; buffer B: 30% methanol in 50 mM potassium phosphate buffer, pH 7.0, supplemented with 10 mM tributylammonium hydroxide). Compound identification was achieved by comparison with injected standards. Relative quantification was obtained by peak height measurements in the chromatogram (UV absorbance at 271 nm) in relation to standards. Specific activities of either NrdA (*Figure 2a–b*) or NrdB (*Figure 2c–f*) were determined. Specific activities varied between protein preparations. In some cases the data was standardized to activity percent, where 100% was determined as maximum enzyme activity in a specific condition.

From a direct plot of activity versus concentration of effector, the K$_L$ values for binding of effectors to the s-site and the a-site, were calculated in SigmaPlot using the equation:

$$V = V_{max} \times [dNTP]/(K_L + [dNTP])$$

and K$_i$ for non-competitive dATP inhibition at NrdB was calculated in Sigmaplot using the equation:

$$V = V_{max}/(1 + [dNTP]/K_i)$$

## GEMMA analysis

In GEMMA, biomolecules are electrosprayed into gas phase, neutralized to singly charged particles, and the gas phase electrophoretic mobility is measured with a differential mobility analyzer. The mobility of an analyzed particle is proportional to its diameter, which therefore allows for quantitative analysis of the different particle sizes contained in a sample (*Kaufman et al., 1996*). The GEMMA instrumental setup and general procedures were as described previously (*Rofougaran et al., 2008*). NrdA, NrdB and NrdBΔ99 proteins were equilibrated by Sephadex G-25 chromatography into a buffer containing 100 mM ammonium acetate, pH 7.8. In addition, 2 mM DTT was added to the NrdA protein solutions to increase protein stability. Prior to GEMMA analysis, the protein samples were diluted to a concentration of 0.025–0.1 mg/ml in a buffer containing 20 mM ammonium acetate, pH 7.5, 0.005% (v/v) Tween 20, nucleotides (when indicated), and magnesium acetate (equimolar to the total nucleotide concentration), incubated for 5 min at room temperature, centrifuged and applied to the GEMMA instrument. Protein concentrations higher than normally recommended for GEMMA were needed to see the larger oligomeric complexes and the experiments to measure NrdA-NrdB interactions were run at as low flow rate as possible (driven by 1.4 Psi pressure) to minimize false interactions that may appear with elevated protein concentration if the flow-rate recommended by the manufacturer is used (3.7 Psi). Most of our experiments were performed with a flow rate driven by 2 Psi.

## Analytical size exclusion chromatography

Fast protein liquid chromatography on a Superdex 200 PC 3.2/30 column (with a total volume of 2.4 ml) and ÄKTA prime system (GE Healthcare) was performed. The column was equilibrated with SEC buffer containing 50 mM Tris-HCl pH 8, 50 mM KCl, 10% glycerol, 10 mM magnesium acetate, 2 mM DTT and when applicable either 3 mM ATP or 0.1–0.5 mM dATP. 50 µL samples containing NrdA, NrdB or both subunits in the presence or the absence of indicated amounts of nucleotides, were pre-incubated for 10 min in room temperature, centrifuged and applied to the column at a temperature of 7°C with a flow rate of 0.07 ml/min. When nucleotides were added to proteins, they were also included in the buffer at the same concentration to avoid dissociation of nucleotide-induced protein complexes during the run. Varying concentrations of proteins were used in the range of 10–113 µM and 5–150 µM for NrdA and NrdB respectively. For complex formation, 10–20 µM NrdA and 10–40 µM NrdB were used in ratios of 1:1 or 1:2. Representative SEC chromatograms in which 20 µM NrdA, 20 µM NrdB or a mixture of 25 µM and 50 µM NrdA and NrdB respectively are shown in *Figure 5*. Molecular weight was estimated based on a calibration curve, derived from globular protein standards using high molecular weight SEC marker kit (GE Healthcare). Standard

deviations were calculated from at least 3 s experiments. The experiments in *Figure 5—figure supplement 1* were run in a similar way but using a Superdex 200 10/300 GL column (loop size 100 μl and flow rate 0.5 ml/min) with a SEC buffer containing 50 mM Tris-HCl pH 7.6, 150 mM KCl, and 10 mM magnesium chloride. Some experiments with ATP or dATP in the mobile phase were monitored at 290 nm to reduce the absorbance from the nucleotides.

## EPR measurements

Measurements were performed on a Bruker ELEXYS E500 spectrometer using an ER049X SuperX microwave bridge in a Bruker SHQ0601 cavity equipped with an Oxford Instruments continuous flow cryostat and using an ITC 503 temperature controller (Oxford Instruments, Oxford, United Kingdom). Measurement temperatures ranged from 5 to 32 K, using liquid helium as coolant. The spectrometer was controlled by the Xepr software package (Bruker).

## Bioinformatics

RNR protein sequences were collected and scored using HMMER (*Eddy, 2011*) HMM profiles in the RNRdb (http://rnrdb.pfitmap.org). ATP-cones were identified with HMMER and the Pfam ATP-cone HMM profile: PF03477 (*Finn et al., 2010*). Sequences representing the diversity of NrdB were selected by clustering all NrdB sequences from RefSeq at an identity threshold of 70% using VSEARCH (*Rognes et al., 2016*). Sequences were aligned with ProbCons (*Do et al., 2005*) and reliable positions for phylogenetic reconstruction were manually selected. A maximum likelihood phylogeny was estimated with FastTree 2 (*Price et al., 2010*).

## Isothermal titration calorimetry (ITC) measurements

Isothermal titration calorimetry (ITC) experiments were carried out on a MicroCal PEAQ-ITC system (Malvern Instruments Ltd) in a buffer containing 25 mM HEPES (pH 7.65), 150 mM NaCl, and 10% glycerol, 2 mM tris(2-carboxyethyl)phosphine, and 5 mM $MgCl_2$. Measurements were done at 10°C. The initial injection volume was 0.4 μl over a duration of 0.8 s. All subsequent injection volumes were 2–2.5 μl over 4–5 s with a spacing of 150 s between the injections. Data for the initial injection were not considered. For dATP binding analysis, the concentration of NrdB in the cell was 12 μM and dATP in syringe 120 or 140 μM. Reverse titrations were performed with 113 μM NrdB in the syringe and 12 or 30 μM dATP in the cell. For titration of dATP into NrdBΔ99, protein concentration in the cell and dATP concentration in the syringe were 50 μM and 900 μM respectively. For dADP binding analysis, the NrdB concentration in the cell was 20–50 μM and ligand concentrations in the syringe were 500–750 μM. For titration of ATP into NrdB, cell and syringe concentrations were 50 and 1600 μM respectively. The data were analyzed using the built-in one set of sites model of the MicroCal PEAQ-ITC Analysis Software (Malvern Instruments Ltd). A fixed ligand/protein stoichiometry of 2 was used for dATP to NrdB titrations. Standard deviations in thermodynamic parameters, N and $K_d$ were estimated from the fits of three different titrations.

## Crystallization and data collection

The purified NrdB, digested by thrombin to remove the hexahistidine tag (see above), was used for crystallization. The protein at a concentration of 9.6 mg/ml was mixed with 20 mM $MgCl_2$, 2 mM TCEP and 5 mM dATP, incubated for 30 min and used for setting up drops using commercially available screens. An initial crystal hit was obtained by the sitting drop vapor diffusion method with a protein to reservoir volume ratio of 200:200 nL and incubated with a 45 μl reservoir at 20 °C in a Triple Drop UV Polymer Plate (Molecular Dimensions, UK). A Mosquito nanoliter pipetting robot (TTP Labtech, UK) was used to set up drops, which were imaged by the Minstrel HT UV imaging system (Rigaku Corporation, USA) available at the Lund Protein Production Platform (LP3). Crystals were obtained with a reservoir containing 0.2 M $CaCl_2$, 0.1 M Tris pH 8.0% and 20% w/v PEG 600 (condition #57 of the PACT screen). The crystals were then further optimized using an additive screen (Hampton Research, USA) and diffraction quality crystals were obtained within 1 week from a crystallization solution containing an additional 3% 6-aminohexanoic acid. Crystals were picked up directly from the drop without cryoprotectant and data were collected at 100 K using the ID23-1 beamline at the ESRF, Grenoble, France. The crystals are in space group P1 with unit cell dimensions a = 74.0,

b = 90.5, c = 90.9, α = 110.8, β = 99.0, γ = 114.1, containing one NrdB tetramer in the unit cell. The solvent content is 49.5%.

## Structure determination and model building

The diffraction images were integrated using the program XDS (*Kabsch, 2010*) and scaled using the program Aimless (*Evans and Murshudov, 2013*) from the CCP4 package (*Winn et al., 2011*). The structure was solved by molecular replacement (MR) in two steps, using Phaser (*McCoy et al., 2007*). First the structure of NrdBΔ99 was solved to 1.7 Å resolution using the most homologous structure in the PDB, that of NrdF from *Chlamydia trachomatis* (PDB: 1SYY) (*Högbom et al., 2004*). This structure was rebuilt manually in Coot (*Emsley et al., 2010*) and using Buccaneer (*Cowtan, 2006*) then refined to convergence using Refmac5 (*Murshudov et al., 1997*) and Buster (*Bricogne et al., 2016*). Full details of this structure will be presented elsewhere. In the second step, a multi-body molecular replacement search was carried out using four copies of NrdBΔ99 and four copies of the ATP-cone domain from *P. aeruginosa* NrdA (PDB: 5IM3) (*Johansson et al., 2016*) prepared by side chain truncation using Chainsaw (*Stein, 2008*). A single solution was found in which all 8 bodies were placed, with a translation function Z score (TFZ) of 12.2. After rearrangement of the ATP-cones from the MR solution to the N-termini of their respective core domains, the structure was refined using Refmac5. Some automatic building was performed using Buccaneer and manual rebuilding in Coot. Final refinement was done using Buster (*Bricogne et al., 2016*). Automatically-generated non-crystallographic symmetry restraints were used. Geometry was validated using the MolProbity server (*Chen et al., 2010*).

## Metal characterization using anomalous diffraction

To establish the nature of the metal ions in the crystal, an anomalous diffraction experiment was conducted at three wavelengths, one on the near-high-energy side of the Fe K edge (1.72 Å), where both Mn and Fe should have a strong anomalous signal, one near the Mn K edge (1.87 Å), where only Mn should have a strong signal, and one on the low energy side of the Mn edge (1.92 Å), where neither Mn nor Fe should have a strong signal but $Ca^{2+}$ will have a small residual signal (*Figure 6—source data 1*). These data were collected at EMBL station P14 of the PETRA-III synchrotron in Hamburg, Germany. The data at different wavelengths were collected in non-overlapping stripes on the crystal to avoid radiation damage that might affect the relative anomalous signals, which were found to be weak at all wavelengths. The structure was refined against all three datasets to a common resolution of 2.5 Å and the relative heights of peaks in the anomalous difference maps at were used to identify the metal. These are presented in *Figure 6—source data 2*. The refinement was done and anomalous difference maps calculated and analyzed using Buster (*Bricogne et al., 2016*).

## Acknowledgements

We want to thank Jarone Pinhassi and Sabina Arnautovic, Linneaus University, Kalmar, Sweden, for the kind gift of *Leeuwenhoekiella blandensis* DNA, Yuliya Leontyeva, Jan-Olov Persson and Rolf Sundberg, Department of Mathematics, Stockholm University, Sweden, for excellent statistical analysis of ligand binding, Eva Muñoz and Angel Piñeiro, AFFINImeter, Santiago de Compostela, Spain for valuable assistance with analysis of ITC titrations, Hanna Jankevics Jones and John Stenson, Malvern Instruments Ltd, Malvern, UK for highly appreciated help with protein biophysical characterization and Fredrik Tholander, Karolinska Institute, Stockholm, Sweden, for generously providing his HPLC instrument for the enzyme assays. We thank the crystallization facility at Lund Protein Production Platform for crystallization trials and staff at EMBL Hamburg and ESRF beamlines, in particular Gleb Bourenkov, for assistance with data collection. This study was supported by grants from the Swedish Cancer Society (CAN 2016/670 to BMS), the Swedish Research Council (2016–01920 to BMS, 2016–04855 to DTL), the Wenner-Gren Foundations (to BMS) and the Carl Trygger Foundation (to AH). Work in the laboratory of GB is supported by the Swedish Research Council (621–2014-5670), the Swedish Research Council for Environment, Agricultural Sciences and Spatial Planning (213-2014-880) and the European Research Council (714102).

# Additional information

## Funding

| Funder | Grant reference number | Author |
| --- | --- | --- |
| Cancerfonden | CAN 721 2016/670 | Britt-Marie Sjöberg |
| Vetenskapsrådet | 2016-01920,2016-04855,621-2014-5670 | Gustav Berggren Derek Logan Britt-Marie Sjöberg |
| Wenner-Gren Foundation | | Britt-Marie Sjöberg |
| Carl Tryggers Stiftelse för Vetenskaplig Forskning | | Anders Hofer |
| Svenska Forskningsrådet Formas | 213-2014-880 | Gustav Berggren |
| H2020 European Research Council | 714102 | Gustav Berggren |

The funders had no role in study design, data collection and interpretation, or the decision to submit the work for publication.

## Author contributions

Inna Rozman Grinberg, Conceptualization, Data curation, Formal analysis, Investigation, Visualization, Methodology, Writing—original draft, Writing—review and editing; Daniel Lundin, Conceptualization, Formal analysis, Investigation, Visualization, Methodology, Writing—original draft, Writing—review and editing; Mahmudul Hasan, Formal analysis, Investigation, Methodology, Writing—review and editing; Mikael Crona, Formal analysis, Methodology; Venkateswara Rao Jonna, Christoph Loderer, Methodology; Margareta Sahlin, Methodology, Writing—review and editing; Natalia Markova, Data curation, Formal analysis, Funding acquisition; Ilya Borovok, Conceptualization, Writing—review and editing; Gustav Berggren, Investigation, Methodology, Writing—review and editing; Anders Hofer, Conceptualization, Data curation, Formal analysis, Funding acquisition, Investigation, Methodology, Writing—review and editing; Derek T Logan, Conceptualization, Data curation, Formal analysis, Funding acquisition, Visualization, Writing—original draft, Writing—review and editing; Britt-Marie Sjöberg, Conceptualization, Data curation, Formal analysis, Supervision, Funding acquisition, Writing—original draft, Project administration, Writing—review and editing

## Author ORCIDs

Inna Rozman Grinberg (iD) http://orcid.org/0000-0003-3094-1998
Daniel Lundin (iD) http://orcid.org/0000-0002-8779-6464
Mahmudul Hasan (iD) https://orcid.org/0000-0002-1767-6440
Venkateswara Rao Jonna (iD) http://orcid.org/0000-0001-7211-4830
Derek T Logan (iD) https://orcid.org/0000-0002-0098-8560
Britt-Marie Sjöberg (iD) http://orcid.org/0000-0001-5953-3360

## Decision letter and Author response

Decision letter https://doi.org/10.7554/eLife.31529.022
Author response https://doi.org/10.7554/eLife.31529.023

# Additional files

## Supplementary files

• Transparent reporting form
DOI: https://doi.org/10.7554/eLife.31529.020

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
