## [Decision Letter]

Thank you for submitting your article "Unique ATP-cone-driven allosteric regulation of ribonucleotide reductase via the radical-generating subunit" for consideration by *eLife*. Your article has been reviewed by two peer reviewers, and the evaluation has been overseen by a Reviewing Editor and Michael Marletta as the Senior Editor. The following individual involved in review of your submission has agreed to reveal his identity: Christopher K Mathews (Reviewer #3).

The reviewers have discussed the reviews with one another and the Reviewing Editor has drafted this decision to help you prepare a revised submission.

Summary:

In the present work, Grinberg et al. describe the structure and function of a ribonucleotide reductase (RNR) from the bacterium, L. blandensis. RNR is an essential enzyme that plays a key role in DNA synthesis and repair, and is a textbook exemplar for allosteric regulation in enzymes. The current paper is well-written and relies on multiple techniques and contributions from several groups in the RNR field. The most significant finding is that a well-studied region of RNRs called the ATP cone, which comprises the regulatory site for controlling RNR activity and is usually located on the large subunit of the heteromeric complex, is unexpectedly found to reside in the small subunit, which generates the free radical required for enzyme catalysis. This observation represents a substantial paradigm shift in the RNR world.

Essential revisions:

1) In the present work, dADP is shown for the first time to act as an allosteric inhibitor. dADP appears to have half the affinity for the activity site compared to the canonical allosteric inhibitor, dATP. Does this finding imply that dADP has a physiological role in regulation? What is the effect of dADP on oligomerization compared to dATP? The k_on_ for dADP should be compared between L. blandensis RNR and the triphosphate-forming kinases that convert dADP to dATP to see whether such a role is feasible. In light of this unusual finding concerning dADP regulation, it is important to carry out a comparison of the binding sites between the L. blandensis structure and the ligand-bound ATP cones from *E. coli* and/or human RNR. Such an analysis may reveal why dADP is not an allosteric inhibitor in *E. coli* or human RNR when it is in L. blandensis. A comparative ITC experiment would be useful to shed light on dADP binding between species, while an anomalous scattering experiment would help solidify the identity of the metal ions observed in the L. blandensis RNR metal cluster and activity site.

2) Two ATP molecules are found to be bound in the activity site of L. blandensis RNR. What is the role of the second one? Based on early work with the eukaryotic RNRs, it was shown that oligomerization is induced by ATP or dATP binding – what is the driving force for oligomerization here? Please address whether the oligomerization phenomena be validated in this system using site-directed mutants.

3) Some additional explanation is required with the GEMMA analysis. In Figure 4, the NrdBΔ99 mutant appears to form monomers, dimers, and trimers without any effector present. Please explain how this is possible. Is it due to error in the dimer and trimer molecular weights, or is there a problem with the approach? Figure 4 shows NrdA is able to form monomers and dimers in the absence of an effector. Please explain why this is – is dimerization due to mass action? In the second trace, where 50 μM of dATP is added, monomers and dimers both appear; however, the monomer is slightly more populated than the dimer. Please explain. Is there any reason, as with other RNR's (such as eukaryotic or *E. coli*), why the dimer population is not dominant?

4) Several assumptions have been made based on the structure in interpreting the ITC data. For example, the number of dATP binding sites is assumed to be two because that is what is seen crystallographically. This stoichiometry should be dealt in an unbiased way by fitting the data to different numbers of sites and using the fit with least error to select the correct model.

5) Comparison of the ΔH values between the different ligands gives a rather surprising value for dADP, which is at least 24 kJ/mol higher than the dATP. This difference implies that several additional hydrogen bonds, ion-pairs, and/or van der Waals interactions are made by dADP compared to dATP. Yet the structure indicates that dATP makes only a single an additional hydrogen bond compared to the dADP. This discordance between the ITC and structural data should be explained.

6) The significance of the work should be better clarified for a general (non-RNR) audience. Given the wealth of important enzyme systems in biology, why are the current findings important for advancing our understanding of broad issues, such as allosteric regulations, the appropriate control of nucleotide levels to avoid disease states, etc. Along these lines, the discussion of the potential applications of the findings is brief and unsatisfying. It is claimed that the new observations offer a possible route to regulate engineered enzymes by dATP inhibition. Please elaborate on this concept and how it might work.

Reviewer #2:

Grinberg et al. describe the structure and function of L. blandensis this is a well-written paper that have contributions of many groups from Sweden that work on Rnr. The major finding in this work is that the ATP cone where lies the activity site, usually located on the large subunit is now found in the small subunit housing the essential free radical. This was totally unexpected and is an important finding that furthers our understanding of Rnr. However, there are some important questions that are prompted by this work, which requires some answers. For example, dADP for the first time shown to act as an allosteric inhibitor. The dADP has half the affinity for the activity site compared to the allosteric inhibitor dATP. Therefore, does this dADP have a physiological role in regulation? For example, what is its efficiency on oligomerization compared to dATP? An experiment with dNTP pool analysis will be required to show if the ADP has a physiological role to play. One would have to compare the k_on_ for the dADP between Rnr and the triphosphate converting kinases that convert the dADP to dATP to see if this is feasible. In the light of this unusual finding, it is important to do a comparison of the binding sites between the L. blandensis structure and the *E. coli*. or human ligand bound ATP cones. This may reveal why ADP is not an allosteric inhibitor in *E. coli*. or human while it is in L. blandensis. Perhaps a comparative ITC experiment might shed light between species. In anomalous scattering experiment would shed light on the identity of the metal ions both on the metal cluster and activity site.

Two ATP molecules are bound in the activity site. What is the role of the second one? Based on the early work with the eukaryotic RNR's it was shown that the oligomerization is induced by ATP or dATP binding what is the force driving the oligomerization here? Can you validate the oligomerization phenomena in the system using site – directed mutants?

Reviewer #3:

This is a major paper, using multiple benchmark techniques, that explores an unexpected finding-that a newly characterized form of ribonucleotide reductase contains its "activity site," AKA ATP-cone, on the small (radical-containing) subunit. A substantial paradigm shift in the RNR world. I have just three suggestions for improving this outstanding paper.

1) Change "Unique" to "Novel" in the title, inasmuch as several different bacteria have been found to share this unexpected property, not just the species chosen for study in this paper.

2) Define the criteria, either amino acid sequence or 3D structure, by which an amino acid sequence of about 100 residues is identified as an "ATP-cone."

3) Either delete the last sentence in the Abstract or (preferably) modify the discussion appropriately, because the discussion of potential applications of the findings in this paper is brief and unsatisfying.

---

## [Author Response]

Essential revisions:1) In the present work, dADP is shown for the first time to act as an allosteric inhibitor. dADP appears to have half the affinity for the activity site compared to the canonical allosteric inhibitor, dATP. Does this finding imply that dADP has a physiological role in regulation?

Actually, dADP has an affinity which is only a fifteenth of the dATP affinity (Figure 8) and because of that we believe that dATP is the main physiological allosteric effector. However, the relative affinity for dADP compared to dATP is still higher than in most other RNRs and we cannot exclude the possibility that local concentrations around the enzyme might be high enough to actually have a physiological effect. Nevertheless, we have modified the text to make it clear that we believe that dATP is the main physiological effector of the two.

What is the effect of dADP on oligomerization compared to dATP? The k_on_ for dADP should be compared between L. blandensis RNR and the triphosphate-forming kinases that convert dADP to dATP to see whether such a role is feasible.

The main message of our current study is the transfer of an ATP-cone domain from the catalytic subunit of an RNR to its corresponding radical-containing subunit, and that the ATP-cone confers similar effects of dATP and ATP to the oligomeric status of the NrdB subunit. We would therefore prefer not to focus the current study on the binding of dADP to the ATP-cone, as we have shown that the binding of dADP is only a fifteenth compared to binding of dATP. Both dADP and dATP induce the formation of NrdB tetramers but the effect of dADP is weaker. We have replaced the dADP experiment in Figure 4Figure 3 with a more representative one. The main conclusion, that both nucleotides induce the formation of NrdB tetramers, is the same. The isotherms for calculating k_on_ values for dADP association with *L. blandensis* NrdB have a subtle shoulder relating to the fact that the two sites are probably not totally equivalent and the rough estimates are 3x10^3^ s^-1^M^-1^s^-1^, compared to human nucleoside diphosphate kinase with k_on_ values for dADP in the order of 4.5x10^6^ s^-1^M^-1^s^-1^ (Schaertl & al, J Biol Chem 1998, 273: 5662). Kinetic parameters derived from the ITC isotherm were obtained using the AFFINImeter software assuming a 1:1 interaction model, according to the method described in J. Am. Chem. Soc., 2012, 134(1), 559-565 (see eq. 4 in Suppl. Information) and Dumas et al. (2016), Methods in Enzymology, Vol. 567, Chap. 7, A.Feig Ed.

In light of this unusual finding concerning dADP regulation, it is important to carry out a comparison of the binding sites between the L. blandensis structure and the ligand-bound ATP cones from E. coli and/or human RNR. Such an analysis may reveal why dADP is not an allosteric inhibitor in E. coli or human RNR when it is in L. blandensis.

The *L. blandensis* ATP cone belongs to the recently-identified family of ATP cones that bind two molecules of dATP rather than only one, as in the human and *E. coli* enzymes. A detailed comparison of the two types of ATP cone was already made for its occurrence in the NrdA protein from *Pseudomonas aeruginosa* in Johansson et al., 2016. The ability to bind a second dATP molecule is due to the presence of a unique sequence motif in the C-terminal helix of the ATP cone domain that is not present in human or *E. coli* RNR. The interactions are essentially identical in the *L. blandensis* ATP cone, thus such a comparison would be superfluous in this paper. Nevertheless, one can speculate that, since the two dATP molecules bind with their triphosphate tails coordinating a Mg^2+^ ion, the loss of the γ phosphate group in dADP would have a relatively minor effect on affinity, as this group makes only one polar interaction with the protein in one of the two dATP molecules. On the other hand, this appears to be the case for the *E. coli* and human ATP cones as well. We hypothesize that the effect of loss of this single interaction is less when two dATP molecules are bound than when there is only one.

A comparative ITC experiment would be useful to shed light on dADP binding between species, while an anomalous scattering experiment would help solidify the identity of the metal ions observed in the L. blandensis RNR metal cluster and activity site.

Anomalous diffraction experiments have been done at three wavelengths (1.72 Å, 1.87 Å and 1.92 Å) that should help discriminate between Fe, Mn and Ca. The results suggest that the metal centre is filled with Ca^2+^ ions from the crystallization medium. This data was added to the main text. ITC studies on dADP binding to other species are beyond the scope of our current study, as we have determined dADP binding to be more than an order of magnitude worse than dATP binding and in vivo concentrations of dNDP in general are known to be low compared to dNTP concentrations.

2) Two ATP molecules are found to be bound in the activity site of L. blandensis RNR. What is the role of the second one? Based on early work with the eukaryotic RNRs, it was shown that oligomerization is induced by ATP or dATP binding – what is the driving force for oligomerization here? Please address whether the oligomerization phenomena be validated in this system using site-directed mutants.

For the new type of ATP cone observed in this enzyme, this question was already addressed quite extensively from a structural point of view in Johansson et al., 2016, where the structure of such a domain was seen for the first time in NrdA from *P. aeruginosa.* The domain was characterized biochemically in a previous paper from the Hofer lab (Jonna et al., 2015). Site-directed mutants were made in the *P. aeruginosa* NrdA interaction surface that abolished tetramerization. Since the interface residues are conserved in *L. blandensis* NrdB and the interactions between ATP cones are almost identical, we do not think it necessary to re-make these mutations.

Regarding the role of the second dATP molecule, this was also covered in Johansson et al., 2016. Inhibition is driven by dATP binding to both kinds of ATP cones (binding one or two dATPs). This induces conformational changes in the domain that leads them to associate, either with each other (as in yeast, human, *P. aeruginosa* and *L. blandensis*) or with NrdB (as in *E. coli)*. The ATP-induced oligomerization is unique to eukaryotic RNRs and has not been characterized structurally. The *L. blandensis* type of ATP cone domain uses a different surface to oligomerize (building tetramers) than what eukaryotes or *E. coli* do, and interestingly this is the surface closest to the second dATP molecule.

3) Some additional explanation is required with the GEMMA analysis. In Figure 4, the NrdBΔ99 mutant appears to form monomers, dimers, and trimers without any effector present. Please explain how this is possible. Is it due to error in the dimer and trimer molecular weights, or is there a problem with the approach? Figure 4 shows NrdA is able to form monomers and dimers in the absence of an effector. Please explain why this is – is dimerization due to mass action? In the second trace, where 50 μM of dATP is added, monomers and dimers both appear; however, the monomer is slightly more populated than the dimer. Please explain. Is there any reason, as with other RNR's (such as eukaryotic or E. coli), why the dimer population is not dominant?

The NrdBΔ99 mutant is mainly a monomer but with a slight tendency to aggregate non-specifically in the ammonium acetate buffer. A typical pattern of aggregation is a ladder of dimers, trimers, tetramers etc, which you also see in the figure. Another possibility could also be false interactions, but by comparing the analysis at many different flow-rates (different pressures), we found that this is less likely. Generally, the role of tiny peaks (now shown in brackets) should not be overemphasized in GEMMA (the trimers, tetramers etc can be regarded minor in this sense whereas the monomers and dimers are much higher). It is not surprising that the NrdA protein is in a monomer-dimer equilibrium already without effector, since there is a great variation between species. For example, the mammalian protein is a monomer and the *E. coli* protein is in a monomer-dimer equilibrium. The relative amounts of monomers and dimers are dependent on the protein concentration. As the reviewer has pointed out, it is a bit surprising that nucleotide effectors do not have as strong effect on dimerization as in most other species studied. All these points are now explained in the text.

4) Several assumptions have been made based on the structure in interpreting the ITC data. For example, the number of dATP binding sites is assumed to be two because that is what is seen crystallographically. This stoichiometry should be dealt in an unbiased way by fitting the data to different numbers of sites and using the fit with least error to select the correct model.

N values in ITC fits are strongly affected by the content of *active* protein material as opposed to total protein concentration. In all the nucleotide titrations the fitted apparent N value was significantly above 1 when total protein concentration was used. However, for the N value to be meaningful in terms of stoichiometry it has to be an integer. Using N=1 for dATP binding the active protein concentration becomes more than 100% of the total concentration, whereas using N=2 gives an active protein concentration of 60%, in our experience a typical fraction of active RNR (the error of the fit will not change whether the N value or the active protein concentration is fixed). The proportion of inactive protein can be explained by e.g. aggregation as observed by SEC-MALS and DLS. Moreover, our X-ray crystallography shows a stoichiometry of 2 bound dATP molecules. For meaningful results, we therefore used the active protein concentration for calculation of ATP and dADP binding stoichiometries.

5) Comparison of the ΔH values between the different ligands gives a rather surprising value for dADP, which is at least 24 kJ/mol higher than the dATP. This difference implies that several additional hydrogen bonds, ion-pairs, and/or van der Waals interactions are made by dADP compared to dATP. Yet the structure indicates that dATP makes only a single an additional hydrogen bond compared to the dADP. This discordance between the ITC and structural data should be explained.

The overall enthalpy of the reaction, measured by calorimetry, is affected not only by direct protein interaction with the ligand, but also by water molecules in the binding pocket (Klebe, Nature Rev Drug Discovery 2015, 14: 95). Since calorimetry measures the overall effect of all changes in the binding pocket, it is hard to correlate and compare enthalpies without structural data for the dADP bound complex.

6) The significance of the work should be better clarified for a general (non-RNR) audience. Given the wealth of important enzyme systems in biology, why are the current findings important for advancing our understanding of broad issues, such as allosteric regulations, the appropriate control of nucleotide levels to avoid disease states, etc. Along these lines, the discussion of the potential applications of the findings is brief and unsatisfying. It is claimed that the new observations offer a possible route to regulate engineered enzymes by dATP inhibition. Please elaborate on this concept and how it might work.

We have added a sentence to the Abstract, a paragraph to the Discussion, plus a few references in the Introduction, describing the degree of novelty of our finding. That allosteric regulation controlled by modular domains is evolutionarily dynamic has been shown before (by us and others), but we are not aware of other enzyme complexes in which a domain has transferred to a non-homologous component. We would like to argue that this, in addition to the general importance of ribonucleotide reduction and its control in all organisms, makes our finding interesting to a broad audience of biologists interested in evolution of proteins, nucleotide metabolism and enzyme structure and function. In addition, we have added a few sentences at the end of the fourth paragraph in the Introduction describing briefly that RNRs are potential targets both for antibacterial and antitumor therapies. We have also added a sentence to the Introduction describing in some more detail how we foresee that ATP-cones could be used to confer regulation of engineered enzymes by control of nucleotide concentrations, at least for enzymes that are sensitive to oligomerization states. Since this is not within our direct area of expertise, we fear that further detail would become too speculative.